# DIFFERENTIALLY PRIVATE SGD WITHOUT CLIPPING BIAS: AN ERROR-FEEDBACK APPROACH

**Xinwei Zhang**
University of Minnesota
zhan6234@umn.edu

**Zhiqi Bu**
Amazon AI.
woodyx218@gmail.com

**Zhiwei Steven Wu**
Carnegie Mellon University
zstevenwu@cmu.edu

**Mingyi Hong**
University of Minnesota
mhong@umn.edu

## ABSTRACT

Differentially Private Stochastic Gradient Descent with Gradient Clipping (DPSGD-GC) is a powerful tool for training deep learning models using sensitive data, providing both a solid theoretical privacy guarantee and high efficiency. However, using DPSGD-GC to ensure Differential Privacy (DP) comes at the cost of model performance degradation due to DP noise injection and gradient clipping. Existing research has extensively analyzed the theoretical convergence of DPSGD-GC, and has shown that it only converges when using *large* clipping thresholds that are dependent on problem-specific parameters. Unfortunately, these parameters are often unknown in practice, making it hard to choose the optimal clipping threshold. Therefore, in practice, DPSGD-GC suffers from degraded performance due to the *constant* bias introduced by the clipping. In our work, we propose a new error-feedback (EF) DP algorithm as an alternative to DPSGD-GC, which not only offers a diminishing utility bound *without* inducing a constant clipping bias, but more importantly, it allows for an *arbitrary* choice of clipping threshold that is independent of the problem. We establish an algorithm-specific DP analysis for our proposed algorithm, providing privacy guarantees based on Rényi DP. Additionally, we demonstrate that under mild conditions, our algorithm can achieve nearly the same utility bound as DPSGD without gradient clipping. Our empirical results on standard datasets show that the proposed algorithm achieves higher accuracies than DPSGD while maintaining the same level of DP guarantee.

## 1 INTRODUCTION

**Background.** Deep learning models have demonstrated exceptional promise in understanding various types of data, including images, texts, speech, and others. The exploding data volume has significantly accelerated the development of deep learning and has led to remarkable success in various tasks, including computer vision (Dosovitskiy et al., 2020), natural language processing (Vaswani et al., 2017), and speech recognition (Gulati et al., 2020). However, recent research (Nasr et al., 2018; Zhu et al., 2019) has shown that the training and inference processes of deep learning models may leak sensitive information in the training data, such as typing history, financial records, medical records, and social network data. To address this concern, the concept of differential privacy (DP) introduced by Dwork (2006) has become a widely accepted privacy requirement for releasing datasets (Dwork, 2008; Wang et al., 2016) and training machine learning models (Bassily et al., 2014; Abadi et al., 2016; Wang et al., 2020; Chen et al., 2020). The DP notion provides a quantitative measurement that reflects the abstract privacy requirement in a general setting. Intuitively, DP prevents adversarial third parties from identifying whether any piece of data has appeared in the dataset or has been used for training the model, with access to all released information. The notion of DP has also been integrated into the procedure of training deep learning models, such as DPSGD (Abadi et al., 2016) in centralized training and DP-FedAvg (Andrew et al., 2021; McMahan et al., 2018b) in distributed optimization.

The DP guarantee of DPSGD relies on injecting DP noises into the released updates at each iteration, and the variance of the injected noise depends crucially on the *sensitivity* of the algorithm. In the practical implementation of DP-SGD, the *gradient clipping* operation is used for bounding the algorithm sensitivity of each update in DPSGD (Abadi et al., 2016). Although enjoying a promising theoretical privacy guarantee and simple implementation, the DPSGD algorithm with gradient clipping (DPSGD-GC) still faces critical challenges in theoretical analysis and practical implementation.

**Challenges.** In terms of theory, although the inclusion of clipping operation in DPSGD-GC ensures a strong DP guarantee, it considerably complicates the convergence analysis compared to the vanilla SGD algorithm. This is because the expected update direction, which is the expected clipped per-sample gradient in DPSGD-GC, may change dramatically, and additional effort is required to analyze its alignment with the true gradient. Therefore, the early works on DPSGD with convergence analysis assume that the clipping threshold is chosen to be larger than the magnitude of each per-sample gradient, essentially making the clipping operation *ineffective* during training (Bassily et al., 2014; Wang et al., 2016; Feldman et al., 2020; Iyengar et al., 2019; Xu et al., 2021; Zhang et al., 2022; Li et al., 2022). Recent works use alternative assumptions and improve the convergence analysis for DPSGD-GC, but the convergence results still rely on an assumption-dependent choice of the clipping threshold (Fang et al., 2022; Chen et al., 2020; Yang et al., 2022; Qian et al., 2021; Zhang et al., 2020; Koloskova et al., 2023). However, the bounds in the assumptions of real-world problems are hard to estimate, and such a choice of clipping threshold is *impossible* to be satisfied in practice. Recent work Koloskova et al. (2023) has shown a negative result that, under the general assumptions for SGD, regardless of the choice of clipping threshold and stepsize, DPSGD-GC converges with a *constant* bias term, meaning in the limit the DPSGD-GC algorithm only converges to a *neighborhood* of the optimal or stationary solution. References Chen et al. (2020); Song et al. (2013) also provide a justification that the gradient clipping shifts the stationary solution of the original problem, thus causing an unavoidable constant bias (see our fixed-point analysis in Section 2.2).

In terms of practical implementation, empirical studies have shown that DPSGD-GC suffers from a severe accuracy drop compared with its non-private counterparts (Abadi et al., 2016; Bagdasaryan et al., 2019; Zhang et al., 2022). The additional terms consist of the bias caused by gradient clipping (as mentioned in the previous paragraph), as well as the term caused by the injected DP noise. It follows that when implementing DPSGD-GC in practice, one often has to carefully tune the clipping threshold so to balance between these two terms. If a small clipping threshold is chosen, DPSGD-GC injects small DP noise into the system, leading to a small DP error term, but at the cost of increased clipping bias. On the other hand, choosing a large clipping threshold reduces the clipping bias, but to ensure the desired DP guarantees, a large DP-noise has to be injected, leading to a large performance drop. Therefore, how to properly choose the clipping threshold in practice is more of an art than a science. Recently, more advanced clipping operations have been used to improve the empirical performance of DPSGD-GC, including adaptive clipping threshold (Andrew et al., 2021), group clipping (McMahan et al., 2018a), micro-batch clipping (Lee et al., 2021), and gradient normalization (Yang et al., 2022; Das et al., 2021). However, the theoretical properties of these approaches are less understood. Additionally, these approaches either entail a trade-off in terms of a weaker DP guarantee or necessitate a substantial amount of parameter tuning.

In summary, extensive research has shown that DPSGD-GC only converges when the clipping thresholds are tuned based on constants appear in various assumptions (such as the magnitude of the gradients (Bassily et al., 2014; Wang et al., 2016; Feldman et al., 2020; Iyengar et al., 2019; Xu et al., 2021; Zhang et al., 2022; Li et al., 2022), the coefficient of the gradient symmetricity (Chen et al., 2020), or per-sample gradient alignment angles (Qian et al., 2021)). Unfortunately, the thresholds are difficult to choose in practice because the aforementioned assumptions are hard to verify, thus the coefficients are typically unknown. Therefore, DPSGD-GC often suffers from degraded performance due to the constant bias introduced by the clipping. This fact strongly motivates a new class of DP algorithms that enjoys *both* DP guarantee without performance degradation, while being free of clipping threshold tuning.

**Our Contributions.** In this work, we propose DiceSGD algorithm for DP training with both utility and privacy guarantees using a problem-independent clipping threshold. DiceSGD is motivated by the error-feedback (EF) mechanism – a classical procedure in signal processing (Howze & Bhattacharyya, 1997; Laakso & Hartimo, 1992) for cancelling quantization bias. Specifically, we propose a novel *clipped* EF mechanism which accumulates the error between the clipped update to the unclipped

one at each iteration, and feeds the *clipped* error back to the next update. The proposed clipped EF mechanism satisfies the DP guarantee, while still preserving the ability to compensate for the per-sample gradient clipping bias and eventually eliminating the convergence bias caused by clipping. In contrast to existing works, the proposed DiceSGD provides DP guarantee and convergence guarantee without constant bias, while allowing a flexible choice of the clipping threshold. More importantly, we have observed that when the algorithm is applied to a number of applications, including image classification and natural language processing tasks, it does not suffer from performance degradation; nor does it require careful clipping threshold tuning.

We emphasize that the theoretical analysis for the proposed DiceSGD is challenging in the following sense: the clipping operation does not satisfy the firmly contracting assumption used in the typical analysis of EF algorithms; additionally, directly applying the conventional DP analysis to DiceSGD leads to an extremely loose bound. Therefore, a new convergence and privacy analysis for the designed algorithm is required. We summarize our major contribution as follows:

- We propose a novel DiceSGD algorithm, where a new *clipped EF mechanism* is designed to eliminate the clipping bias, while still providing the algorithm with standard DP guarantee.
- We provide the convergence proof for DiceSGD under general non-convex and Lipschitz-smooth assumption, and show that DiceSGD eliminates the constant clipping bias compared with DPSGD-GC with an arbitrary constant clipping threshold.
- We develop an *algorithm-specific* Rényi-DP analysis for the proposed method, where the update consists of a privatized state and a *non-privatized hidden* state. We show that DiceSGD satisfies $(\epsilon, \delta)$-DP by injecting a slightly (i.e., a constant depending on the clipping threshold of the feedback error signal) larger DP noise compared with DPSGD-GC.
- Finally, we perform rigorous empirical comparisons of our method to DPSGD-GC on a number of publicly available datasets to demonstrate the ability of our method to train models with a high privacy guarantee and good performance. Further, we conduct ablation studies on DiceSGD to show its stability in the choice of hyper-parameters.

## 2 PRELIMINARIES

### 2.1 NOTATIONS AND ASSUMPTIONS

**Problem formulation** Throughout the paper, we consider the following empirical risk minimization (ERM) problem on a dataset $\mathcal{D} := \{\xi_i, i \in [1, \ldots, N]\}$ consisting of $N$ samples of $\xi_i$:

$$\min_{\mathbf{x} \in \mathbb{R}^d} f(\mathbf{x}) := \frac{1}{N} \sum_{\xi \in \mathcal{D}} f(\mathbf{x}; \xi), \tag{1}$$

where $\mathbf{x} \in \mathbb{R}^d$ denotes the model parameter of dimension $d$. Further, we denote the per-sample gradient evaluated at $\mathbf{x}^t$ and sample $\xi_i$ as $\mathbf{g}_i^t = \nabla f(\mathbf{x}^t, \xi_i)$. The clipping operation applied to vector $\mathbf{v}$ is defined as:

$$\text{clip}(\mathbf{v}, C) = \min \left\{ 1, \frac{C}{\|\mathbf{v}\|} \right\} \cdot \mathbf{v}. \tag{2}$$

Throughout the paper, we use superscript $(\cdot)^t$ to denote the variables in iteration $t$, and $\mathcal{B}$ to denote the index set of the sampled minibatch from dataset $\mathcal{D}$. The formal definition of differential privacy (DP) is stated below:

**Definition 2.1** ($\epsilon, \delta$-DP (Dwork, 2006)). A randomized mechanism $\mathcal{M}$ is said to guarantee $(\epsilon, \delta)$-differentially private, if for any two neighboring datasets $\mathcal{D}, \mathcal{D}'$ ($\mathcal{D}, \mathcal{D}'$ differ by one sample instance) and for any output measurement $\mathcal{S}$, it holds that $\Pr[\mathcal{M}(\mathcal{D}) \in \mathcal{S}] \leq e^\epsilon \Pr[\mathcal{M}(\mathcal{D}') \in \mathcal{S}] + \delta$.

To protect DP, we consider the commonly used Gaussian mechanism (Dwork, 2006; Abadi et al., 2016), which injects additive noise into the output of the algorithm.

**Definition 2.2** (Gaussian Mechanism (Dwork, 2006)). Suppose an algorithm $f : \mathcal{D} \to \mathbb{R}^d$ has $\ell_2$ sensitivity $\Delta_f$

$$\max_{\mathcal{D}, \mathcal{D}'} \|f(\mathcal{D}) - f(\mathcal{D}')\| \leq \Delta_f.$$

Then for any $\epsilon > 0, \delta \leq 1$, by adding a random Gaussian noise to the output of the algorithm $M(x) = f(x) + \mathbf{w}$, with $\mathbf{w} \sim \mathcal{N}(0, \sigma^2 I_d)$, where $\sigma = \frac{\Delta_f \sqrt{2 \ln(1.25/\delta)}}{\epsilon}$, the algorithm $f$ is $(\epsilon, \delta)$-DP.

---
**Algorithm 1** DPSGD Algorithm with Gradient Clipping

---
1: **Input:** $\mathbf{x}^0, \mathcal{D}, C, \eta$
2: **for** $t = 0, \ldots, T-1$ **do**
3:     Uniformly draw minibatch $\mathcal{B}^t$ from $\mathcal{D}$
4:     $\tilde{\mathbf{g}}_i^t = \text{clip}\left(\nabla f(\mathbf{x}^t; \xi_i), C\right)$
5:     $\mathbf{x}^{t+1} = \mathbf{x}^t - \frac{\eta^t}{B}\left(\sum_{i \in \mathcal{B}^t} \tilde{\mathbf{g}}_i^t + \mathbf{w}^t\right),$
6:         where $\mathbf{w}^t \sim \mathcal{N}(0, \sigma_1^2 \cdot \mathbf{I})$
7: **end for**

---

## 2.2 DPSGD-GC ALGORITHM

The update of DPSGD-GC algorithm (Abadi et al., 2016) is given in Algorithm 1. The algorithm first samples a mini-batch $\mathcal{B}^t$ of size $B$ and computes the per-sample gradient at each step. Then, it applies the Gaussian mechanism by clipping the per-sample gradient with (2) and injecting the DP noise. Finally, the algorithm updates the model parameter with the averaged privatized mini-batch gradient. It has been shown that DPSGD-GC guarantees $(\epsilon, \delta)$-DP with sufficiently large injected noise (Abadi et al., 2016).

**Theorem 2.3** (Theorem 1 Abadi et al. (2016)). *Given $N, B, T$ and $C$, there exist positive constants $u, v$, such that for any $\epsilon < \frac{uB^2 T}{N^2}, \delta > 0$, by choosing $\sigma_1^2 \geq v\frac{C^2 T \ln(\frac{1}{\delta})}{N^2 \epsilon^2}$, Algorithm 1 is guaranteed to be $(\epsilon, \delta)$-DP.*

Although providing a strong DP guarantee, the convergence property of DGSGD-GC is less satisfactory. Recent work Koloskova et al. (2023) has shown that without any extra assumption, DPSGD-GC with an *arbitrary* clipping threshold converges with a *constant* clipping bias, regardless of the convexity of the problem. Prior works that show the convergence of DPSGD-GC rely on extra assumptions on the problem and clipping thresholds that depend on these assumptions. Specifically, Chen et al. (2020) proves the convergence of DPSGD-GC under the assumption that the per-sample gradients have a symmetric distribution; Jin et al. (2022) gives a high probability convergence result assuming that the per-sample gradients have a bounded domain and sufficiently large clipping threshold; Yang et al. (2022) establishes the convergence of DPSGD-GC by assuming that the deviation of per-sample gradient from the true gradient is bounded, and using a clipping threshold larger than the per-sample gradient deviation to ensure that clipped gradient "aligns" with the true gradient; light-tailed gradient variance assumption and a large clipping threshold has been used by Fang et al. (2022) to provide a high probability bound without constant bias.

**Fixed-point analysis** To intuitively understand why DPSGD-GC requires additional assumptions on the per-sample gradients and large clipping threshold, let us consider the fixed-point of DPSGD-GC. From the algorithm's update in Algorithm 1, at the fixed point of DPSGD-GC, we have:

$$\mathbb{E}[\mathbf{x}] = \mathbb{E}\left[\mathbf{x} - \frac{\eta}{B}\left(\sum_{i \in \mathcal{B}} \text{clip}\left(\nabla f(\mathbf{x}; \xi_i), C\right) + \mathbf{w}\right)\right] = \mathbb{E}[\mathbf{x}] - \frac{\eta}{N}\sum_{i=1}^N \text{clip}\left(\nabla f(\mathbf{x}; \xi_i), C\right).$$

It indicates that $\frac{1}{N}\sum_{i=1}^N \text{clip}\left(\nabla f(\mathbf{x}; \xi_i), C\right) = 0$ is the fixed-point of DPSGD-GC, but it is clear that such an equality does not imply $\nabla f(\mathbf{x}) = 0$ in general. Thus DPSGD-GC is *not guaranteed* to converge to the solution of the problem (1) where $\nabla f(\mathbf{x}) = 0$. Additionally, from the fixed-point of DPSGD-GC, we can also understand how the extra assumptions and clipping thresholds guarantee convergence. For example, by using a clipping threshold larger than the deviation of per-sample gradient (Yang et al., 2022), it guarantees that when $\nabla f(\mathbf{x}) = 0$, it holds that $\|\nabla f(\mathbf{x}; \xi_i) - \nabla f(\mathbf{x})\| = \|\nabla f(\mathbf{x}; \xi_i)\| \leq C$, and

$$\frac{1}{N}\sum_{i=1}^N \text{clip}\left(\nabla f(\mathbf{x}; \xi_i), C\right) = \frac{1}{N}\sum_{i=1}^N \nabla f(\mathbf{x}; \xi_i) = \nabla f(\mathbf{x}) = 0,$$

becomes the fixed-point of DPSGD-GC.

Although providing theoretically sound convergence analyses, the theoretical results in Chen et al. (2020); Jin et al. (2022); Yang et al. (2022); Fang et al. (2022) do not provide practical guidance on choosing the clipping threshold in real-world applications. In these works, the choices of clipping

thresholds depend on the problem parameters, which are hard or impossible to estimate. Therefore, these analyses cannot guarantee that clipping thresholds used in real-world training satisfy the requirements. Thus, DPSGD-GC still suffers from a constant clipping bias, and there is a strong need to design a new DP algorithm that does not suffer from clipping bias.

### 2.3 ERROR-FEEDBACK (EF) SGD

The EF mechanism has been used to debias the quantization error in signal processing (Laakso & Hartimo, 1992) and has been introduced to optimization algorithms for bias compensation when transmitting biased compressed gradients (Karimireddy et al., 2019; Stich & Karimireddy, 2020; Li et al., 2022). The EF mechanism for compressed SGD (EFSGD) writes (Karimireddy et al., 2019)

$$\mathbf{x}^{t+1} = \mathbf{x}^t - \eta\mathbf{v}^t,$$
$$\mathbf{e}^{t+1} = \mathbf{e}^t + \mathbf{g}^t - \mathbf{v}^t, \tag{3}$$

where $\mathbf{v} := \mathrm{Compress}(\mathbf{e}^t + \mathbf{g}^t)$ is a biased compressor and $\mathbf{g}^t$ is the (estimated) gradient. By using the EF mechanism, the bias caused by compression can be controlled by the stepsize $\eta$ and fully eliminated, thus providing better convergence performance than the original compressed SGD algorithm. In the recent works Richtárik et al. (2021), a Markov EF mechanism is proposed for simpler implementation and is used for both compression and clipping. However, this EF mechanism fails to deal with stochastic noise in the gradient estimation. EF has also been used in distributed DP algorithm with compression (Li et al., 2022), where the proposed SoteriaFL framework adopts a "shifted compression" mechanism to eliminate the compression bias when transmitting the privatized local updates. Although showing promising potential in dealing with biased updates caused by compression, the existing EF mechanism has not been directly applied to debias the gradient clipping operation; nor has it been used as a component in DP algorithms.

## 3 DIFFERENTIALLY PRIVATE CLIPPING ERROR-FEEDBACK SGD

In this section, we present the proposed **Di**fferentially Private **C**lipping **E**rror-Feedback SGD (DiceSGD) algorithm inspired by the EF mechanism, which has both convergence and DP guarantee under an *arbitrary* choice of clipping threshold. We show that under mild assumptions, DiceSGD can *fully eliminate* the clipping bias in DPSGD-GC even when a small and *problem-independent* clipping threshold is used.

### 3.1 ALGORITHM DESCRIPTION

Our DiceSGD algorithm is described in Algorithm 2 and Figure 1. At round $t$, the algorithm first computes the update direction $\mathbf{v}^t$ by adding the clipped stochastic gradient with the clipped feedback error. Then, the algorithm updates the model parameters $\mathbf{x}^t$ with $\mathbf{v}^t$ and injects the DP noise $\mathbf{w}^t$. Finally, it computes the clipping error $\mathbf{e}^{t+1}$ for the next iteration. The algorithm only releases $\mathbf{x}^t$ at iteration $t$ and does not release $\mathbf{e}^t$ nor $\mathbf{v}^t$.

In the proposed algorithm, we introduce an extra variable $\mathbf{e}^t$ that records the clipping error. We keep it unclipped and privatize it when computing the update direction in the next iteration. As an important algorithm design consideration for DP requirement, unlike the original EF mechanism, we do not feed $\mathbf{e}^t$ back directly to each per-sample gradient clipping operation (Line 5), because it would break the sensitivity of the algorithm. Rather, we first clip $\mathbf{e}^t$ and add it to the averaged clipped gradient. Using such a clipped EF mechanism for privacy guarantee, we can balance the functionality of EF and the DP requirement of the algorithm.

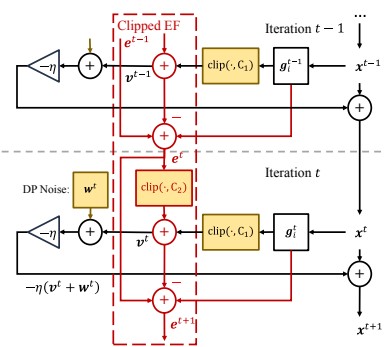

Figure 1: The flow diagram of DiceSGD. The clipped EF components are highlighted in red, and DP components are marked in yellow. $z^{-1}$ denotes the unit delay.

To see why the proposed algorithm has the potential of eliminating the clipping bias, let us again study the fixed-point of the DiceSGD algorithm. At the fixed-point, we have the following relation, where the expectation $\mathbb{E}[\cdot]$ is

---

**Algorithm 2** DiceSGD Algorithm

1: **Input:** $\mathbf{x}^0, \mathcal{D}, C_1, C_2, \eta$
2: **Initialize:** $\mathbf{e}^0 = 0$
3: **for** $t = 0, \ldots, T - 1$ **do**
4:      Randomly draw minibatch $\mathcal{B}^t$ from $\mathcal{D}$
5:      $\mathbf{v}^t = \frac{1}{B} \sum_{i \in \mathcal{B}^t} \text{clip} \left( \nabla f(\mathbf{x}^t; \xi_i), C_1 \right) + \text{clip} \left( \mathbf{e}^t, C_2 \right)$
6:      $\mathbf{x}^{t+1} = \mathbf{x}^t - \eta^t (\mathbf{v}^t + \mathbf{w}^t)$, where $\mathbf{w}^t \sim \mathcal{N}(0, \sigma_1^2 \cdot \mathbf{I})$
7:      $\mathbf{e}^{t+1} = \mathbf{e}^t + \frac{1}{B} \sum_{i \in \mathcal{B}^t} \nabla f(\mathbf{x}^t; \xi_i) - \mathbf{v}^t$.
8: **end for**

---

taken on the randomness of the samples at the current iteration.

$$\mathbb{E}[\mathbf{x}] = \mathbb{E}[\mathbf{x}] - \eta \, \mathbb{E}[\mathbf{v} + \mathbf{w}] = \mathbf{x} - \eta \, \mathbb{E}[\mathbf{v}],$$

$$\mathbb{E}[\mathbf{e}] = \mathbb{E}[\mathbf{e}] + \mathbb{E}[\frac{1}{B} \sum_{i \in \mathcal{B}} \nabla f(\mathbf{x}; \xi_i) - \mathbf{v}] = \mathbb{E}[\mathbf{e}] + \frac{1}{N} \sum_{i=1}^N \nabla f(\mathbf{x}; \xi_i) - \mathbb{E}[\mathbf{v}].$$

Therefore, from the above two equations, we can derive that

$$\mathbb{E}[\mathbf{v}] = \frac{1}{N} \sum_{i=1}^N \text{clip} \left( \nabla f(\mathbf{x}; \xi_i), C_1 \right) + \text{clip} \left( \mathbf{e}, C_2 \right) = 0,$$

$$\frac{1}{N} \sum_{i=1}^N \text{clip} \left( \nabla f(\mathbf{x}; \xi_i), C_1 \right) + \text{clip} \left( \mathbf{e}, C_2 \right) = \frac{1}{N} \sum_{i=1}^N \nabla f(\mathbf{x}; \xi_i),$$

which indicates that the fixed-point of DiceSGD is given by

$$\frac{1}{N} \sum_{i=1}^N \text{clip} \left( \nabla f(\mathbf{x}; \xi_i), C_1 \right) = -\text{clip} \left( \mathbf{e}, C_2 \right), \ \text{ and } \ \frac{1}{N} \sum_{i=1}^N \nabla f(\mathbf{x}; \xi_i) = \nabla f(\mathbf{x}) = 0.$$

We can show that when $C_2 \geq C_1$, there *exists* $\mathbf{x}, \mathbf{e}$ such that the fixed point is achieved. Specifically, by choosing $\mathbf{x}$ satisfies $\nabla f(\mathbf{x}) = 0$; and $\mathbf{e}$ is chosen as $\mathbf{e} = -\frac{1}{N} \sum_{i=1}^N \text{clip} \left( \nabla f(\mathbf{x}; \xi_i), C_1 \right)$. Note that

$$\|\mathbf{e}\| = \left\| \frac{1}{N} \sum_{i=1}^N \text{clip} \left( \nabla f(\mathbf{x}; \xi_i), C_1 \right) \right\| \leq \frac{1}{N} \sum_{i=1}^N \|\text{clip} \left( \nabla f(\mathbf{x}; \xi_i), C_1 \right)\| \leq C_1,$$

we have $\text{clip} \left( \mathbf{e}, C_2 \right) = \mathbf{e}$ as long as $C_2 \geq C_1$, and the first equation is satisfied. These choices guarantee that the two equations are satisfied and the fixed point is achieved. The fixed-point analysis indicates that, unlike DPSGD-GC, as long as $C_2 \geq C_1$, a condition that is *problem independent*, clipped EF can potentially fully compensate the shift of the stationary solution caused by gradient clipping *independent* of any problem assumptions, and $\nabla f(\mathbf{x}) = 0$ is the fixed point of DiceSGD.

## 3.2 THEORETICAL ANALYSIS

In this section, we provide analysis for the proposed DiceSGD algorithm. We emphasize again that the challenge here is two-fold: 1) it is difficult to analyze convergence due to the combination of the EF mechanism and the clipping operation; 2) the DP analysis is non-trivial due to the presence of the non-privatized update of $\mathbf{e}^t$ as a hidden state. To see the first challenge, more specifically, the analyses of the convergence of the existing EF algorithms (Karimireddy et al., 2019; Li et al., 2022) relies on the assumption that the feedback error $\mathbf{e}^{t+1}$ in (3) is a firmly contractive mapping on $\mathbf{e}^t + \mathbf{g}^t$:

$$\mathbb{E} \left\| \mathbf{e}^{t+1} \right\|^2 = \mathbb{E} \left\| \mathbf{e}^t + \mathbf{g}^t - \text{Compress}(\mathbf{e}^t + \mathbf{g}^t) \right\|^2 \leq \alpha \left\| \mathbf{e}^t + \mathbf{g}^t \right\|^2,$$

where $\alpha \in (0, 1)$ is a constant strictly less than 1. However, in DiceSGD, the clipping error does not satisfy this property. To see this, note the following:

$$\left\| \mathbf{e}^{t+1} \right\|^2 = \left\| \mathbf{e}^t + \frac{1}{B} \sum_{i \in \mathcal{B}^t} \nabla f(\mathbf{x}^t; \xi_i) - \left( \text{clip} \left( \mathbf{e}^t \right) + \frac{1}{B} \sum_{i \in \mathcal{B}^t} \text{clip} \left( \nabla f(\mathbf{x}^t; \xi_i) \right) \right) \right\|^2$$

$$\leq \alpha \left\| \mathbf{e}^t + \frac{1}{B} \sum_{i \in \mathcal{B}^t} \nabla f(\mathbf{x}^t; \xi_i) \right\|^2, \quad \alpha \in (0, 1],$$

which is *non-expansive*, i.e., $\alpha \to 1$ when $\left\| \mathbf{e}^t + \frac{1}{B} \sum_{i \in \mathcal{B}^t} \nabla f(\mathbf{x}^t; \xi_i) \right\| \to \infty$. Therefore, the existing convergence analyses for the EF algorithms cannot be directly applied to our case. On the other hand, privacy analysis for DPSGD is provided in Abadi et al. (2016), where the sequential updates are released, and recent works studying the privacy amplification by iteration provide last-iterate DP analyses for DP algorithms where only the final state is released to public (Feldman et al., 2018; Ye & Shokri, 2022). However, the update of DiceSGD is more complicated than the above two cases, as the sequential update of $\mathbf{x}^t$ is released and privatized, while $\mathbf{e}^t$, the hidden-state with non-privatized updates, is not released to the public. It is insufficient to directly use the existing DP analyses for DiceSGD, because when applying the privacy analysis for DPSGD to the sequence $\{(\mathbf{x}^t, \mathbf{e}^t)\}$ in DiceSGD, the composition theorem does work as $\mathbf{e}^t$ is not privatized. To tackle the above difficulties, we conduct novel analyses for DiceSGD, which consists of the convergence analysis for clipped EF and DP analysis for algorithms with a privatized public state and a non-privatized hidden state.

**Assumptions** We briefly discuss the assumptions used in the analyses of DiceSGD algorithm:

**Assumption 3.1** (Lower Bounded). The loss function $f(\cdot)$ is bounded from below by some finite constant $f^\star$:
$$f(\mathbf{x}) \geq f^\star > -\infty, \ \forall \, \mathbf{x} \in \mathbb{R}^d.$$

**Assumption 3.2** (Smoothness). The loss function $f(\cdot)$ is $L$-Lipschitz smooth, i.e., it satisfies:
$$\|\nabla f(\mathbf{x}) - \nabla f(\mathbf{y})\| \leq L \|\mathbf{x} - \mathbf{y}\|, \forall \, \mathbf{x}, \mathbf{y} \in \mathbb{R}^d.$$

**Assumption 3.3** (Strong Convexity). The loss function $f(\cdot)$ is $\mu$-strongly convex:
$$f(\mathbf{y}) \geq f(\mathbf{x}) + \langle \nabla f(\mathbf{x}), \mathbf{y} - \mathbf{x} \rangle + \frac{\mu}{2} \|\mathbf{x} - \mathbf{y}\|^2, \forall \, \mathbf{x}, \mathbf{y} \in \mathbb{R}^d.$$

Assumptions 3.1 and 3.2 are standard assumptions used for analyzing the convergence of first-order optimization algorithms. The strong convexity assumption has also been widely used in analyzing SGD-type algorithms in both private (Wang et al., 2020; Song et al., 2020; Kamath et al., 2022; Koloskova et al., 2023) and non-private (Rakhlin et al., 2011) settings.

**Assumption 3.4** (Bounded Variance). The stochastic gradient estimation is unbiased, i.e., $\mathbb{E}[\mathbf{g}] = \nabla f(\mathbf{x})$, and its variance satisfies that there exists a constant $\sigma$, such that $\mathbb{E} \|\nabla f(\mathbf{x}) - \mathbf{g}_i\|^2 \leq \frac{\sigma^2}{N}, \forall \, \mathbf{x} \in \mathbb{R}^d$.

**Assumption 3.5** (Bounded Gradient). The gradient of the function is bounded in the sense that there exists a positive constant $G = \sup_{\mathbf{x} \in \mathbb{R}^d} \|\nabla f(\mathbf{x})\| < \infty$.

Assumptions 3.4 and 3.5 are commonly used for analyzing clipping operation (Zhang et al., 2020; Qian et al., 2021; Song et al., 2020), the convergence of DP algorithms (Yang et al., 2022), and distributed optimization (Li et al., 2022; Zhang et al., 2022). Assumption 3.4 assumes a smaller variance compared with the typical assumption (i.e., $\mathbb{E} \|\nabla f(\mathbf{x}) - \mathbf{g}_i\|^2 \leq \sigma^2$), it implies that $\|\nabla f(\mathbf{x}) - \mathbf{g}_i\|^2 \leq \sigma^2, \forall \, i$, and it is necessary for bounding the clipping bias in the existing works (e.g.,in Yang et al. (2022)). Although these assumptions are also used in our analysis, contrasting with existing works, the clipping thresholds $C_1, C_2$ in DiceSGD do not depend on $G$ or $\sigma$.

We now present the convergence theorem of the proposed DiceSGD algorithm under the non-convex smooth setting Assumption 3.2:

**Theorem 3.6.** *Assume the problem satisfies Assumption 3.1, 3.2, 3.4, and 3.5. Given any constant DP noise multiplier $\sigma_1$, by running DiceSGD (Algorithm 2) for $T$ iterations, choosing stepsize $\eta = \sqrt{\frac{2(f(\mathbf{x}^0) - f^\star)}{TL(2C_1^2 + 3C_2^2 + d\sigma_1^2)}}$, clipping thresholds $C_2 \geq 3C_1 + \frac{\sigma}{B} > 0$. It satisfies*

$$\mathbb{E}_t \left[ \|\nabla f(\mathbf{x}^t)\|^2 \right] \leq 2 \sqrt{\frac{2L(f(\mathbf{x}^0) - f^\star)(2C_1^2 + 3C_2^2 + d\sigma_1^2)}{T}}, \tag{4}$$

*where the expectation $\mathbb{E}_t$ is taken over $t \in \{0, \ldots, T - 1\}$, following distribution $\frac{A_t}{\sum_{t=0}^{T-1} A_t}$, with $\{A_t\} \in (0, 1]$ being a strictly positive sequence defined in (12), Appendix A.*

Table 1: The comparison between DPSGD, DPSGD-GC, and DiceSGDin terms of convergence, privacy noise, and clipping thresholds. ($\tilde{G} = 2C^2 + C_1^2$)

| Algorithm | Convergence Rate | Privacy Noise Variance | Assumptions | Clipping |
|---|---|---|---|---|
| DPSGD | $\mathcal{O}\left(\frac{G\sqrt{\log(1/\delta)}}{N\epsilon}\right)$ | $\mathcal{O}(\frac{G\sqrt{T\log(\frac{1}{\delta})}}{N\epsilon})$ | 3.4, 3.5 | $C \geq G + \sigma$ |
| DPSGD-GC | $\mathcal{O}\left(\frac{C\sqrt{\log(1/\delta)}}{N\epsilon}\right) + \mathcal{O}(1)$ | $\mathcal{O}(\frac{C\sqrt{T\log(\frac{1}{\delta})}}{N\epsilon})$ | 3.4, 3.5 | $C < G + \sigma$ |
| **DiceSGD** | $\mathcal{O}\left(\frac{\sqrt{\tilde{G}}\log(1/\delta)}{N\epsilon}\right)$ | $\mathcal{O}(\frac{\sqrt{\tilde{G}T\log(\frac{1}{\delta})}}{N\epsilon})$ | 3.4, 3.5 | Independent of $G$ |

**Proof sketch of Theorem 3.6:**

1. We first apply the convergence analysis of biased SGD for non-convex problems with update direction $\mathbb{E}[\mathbf{v}^t]$. Due to the EF mechanism, the convergence result for DiceSGD directly depends on the recursion of $\mathbf{e}^t$, which corrects the bias at iteration $t-1$.

2. With the update of $\mathbf{e}^t$, we can derive a recursive bound on the key term $\langle \nabla f(\mathbf{x}^t), \mathbb{E}[\mathbf{e}^t] \rangle$. Unlike EF for contracting error, which depends on the gradients with a constant factor independent of $T$, the error $\mathbf{e}^t$ caused by clipping operation requires a much tighter recursion directly on the inner product between $\mathbf{e}^t$ and $\nabla f(\mathbf{x}^t)$ for analysis. And the coefficients before the gradient heavily depend on the clipping factor.

3. By substituting the bound of $\langle \nabla f(\mathbf{x}^t), \mathbb{E}[\mathbf{e}^t] \rangle$ into the convergence result in step 1, and choosing sufficiently small stepsize and adequate clipping factor ratio that compensates for the stochastic noise and the clipping bias, we are able to derive a non-trivial convergence result for DiceSGD.

Theorem 3.6 indicates that the overall convergence rate for DiceSGD is $\mathcal{O}\left(\frac{1}{\sqrt{T}}\right)$ for the general non-convex setting, which matches the $\mathcal{O}(\frac{1}{\sqrt{T}})$ lower bound convergence rate of DPSGD without gradient clipping under non-convexity (Bassily et al., 2014; Rakhlin et al., 2011). However, compared with DPSGD-GC (Koloskova et al., 2023), DiceSGD fully eliminates the constant bias and improves the convergence rate from $\mathcal{O}(1)$ to $\mathcal{O}(\frac{1}{\sqrt{T}})$. The comparison is shown in Table 1.

**Privacy guarantee**   Let us proceed with the privacy analysis of DiceSGD. We start with the notion of Rényi Differential Privacy (Mironov, 2017). By accounting for the distribution divergence of the stochastic gradient at iteration $t$ and the accumulated difference of $\mathbf{e}^t$ starting from $\mathbf{e}^0$, we are able to bound the Rényi divergence of $\mathbf{x}^{t+1}$ given two adjacent datasets $\mathcal{D}, \mathcal{D}'$ and start with the same $\mathbf{x}^t$. Then by using the composition theorem of Rényi divergence, we provide the privacy guarantee for DiceSGD in the next result.

**Theorem 3.7.** *Assume the problem satisfies Assumptions 3.4, and 3.5, given constant $C$, by fixing the clipping thresholds $0 < C_1 \leq C_2 \leq C/B$, independent of $G, \sigma$, and assume $\frac{B}{N} \leq \frac{1}{5}$. Choose DP noise standard deviation $\sigma_1$ as*

$$\sigma_1^2 \geq \frac{32T\tilde{G}\log(1/\delta)}{N^2\epsilon^2},$$

*where $\tilde{G} := C_1^2 + 2\min\{C^2, G'^2\}$, and $G'$ defined in Theorem 3.6. Running DiceSGD for $T$ iteration, the algorithm guarantees $(\epsilon, \delta)$-differentially private.*

Note that although Assumptions 3.4, and 3.5 are used in the proof, the result does not rely on the specific values of the bounds, which can be arbitrarily large. Due to the accumulated influence of the update of $\mathbf{e}^t$, the DiceSGD requires larger DP-noise than the DPSGD algorithm (larger by a constant multiplicative factor). The detailed proof is given in Appendix A.2. By optimizing $T$ we have the following utility-privacy trade-off for DiceSGD.

**Corollary 3.8.** *Under the same assumptions of Theorem 3.6, choose the stepsize $\eta = \mathcal{O}(\frac{1}{\sqrt{T}})$, and clipping thresholds $0 < 3C_1 < C_2 \leq C/B$, and choose noise multiplier $\sigma_1^2$ as Theorem 3.7. By running DiceSGD for $T = \mathcal{O}\left(\frac{N^2\epsilon^2}{\tilde{G}\log(1/\delta)}\right)$ iterations, the algorithm guarantees $(\epsilon, \delta)$-DP, while*

Table 2: Test accuracy of DPSGD-GC and DiceSGD on Cifar-10 and Cifar-100 datasets with different clipping thresholds and $(2, 10^{-5})$-DP.

| Dataset | Clipping. | DPSGD-GC | DiceSGD | SGD |
|---|---|---|---|---|
| Cifar-10 | $C = 1.0$ | 95.2% | 97.4% | 99.0% |
| Cifar-10 | $C = 0.1$ | 94.5% | 97.5% | 99.0% |
| Cifar-100 | $C = 1.0$ | 79.0% | 86.3% | 92.0% |
| Cifar-100 | $C = 0.1$ | 78.9% | 86.5% | 92.0% |

*converging to a solution where the loss function satisfies:*

$$\mathbb{E}[\|\nabla f(\mathbf{x}^t)\|^2] = \mathcal{O}\left(\frac{\sqrt{\tilde{G}\log(1/\delta)}}{N\epsilon}\right).$$

The corollary indicates that when $N \to \infty$, the expected loss converges with rate $\mathcal{O}(\frac{\log(N)}{N^2})$ with arbitrary clipping thresholds $C_2 \geq C_1 > 0$ and eliminates the constant clipping bias in DPSGD-GC.

## 4 NUMERICAL EXPERIMENTS

In the experiment, we use the similar Adam variant of DPSGD-GC developed following Bu et al. (2021) to implement both DPSGD-GC and DiceSGD (see Appendix C.3 for details). We perform extensive evaluations of DiceSGD on image classification, and natural language processing (NLP) tasks to demonstrate its advantage over DPSGD-GC. The experiments were run on an Intel Xeon W-2102 CPU with an NVIDIA TITAN X GPU for image classification, and on an NVIDIA A100 GPU for NLP tasks. We conduct extra ablation studies on the choice of the clipping threshold $C_1, C_2$ and learning rate $\eta$ on Cifar-10 and Cifar-100 datasets, which show that DiceSGD benefits from using a smaller clipping threshold and choosing $C_2 = C_1$ gives the best result in most cases. More results and discussions are given in Appendix C.1 due to the space limitation.

**Image classification.** We use both Cifar-10 and Cifar-100 datasets for experiments and use ViT-small (Dosovitskiy et al., 2020) as the training model, which is pre-trained on Imagenet. We fine-tune the model for 3 epochs with batch size $B = 1000$. The stepsize for DPSGD-GC and DiceSGD are selected through grid search from $\eta \in \{10^{-2}, 10^{-3}, 10^{-4}\}$. The experiment results are shown in Table 2.

**Natural language processing.** To validate the ability of DiceSGD for training larger models on different tasks, we further conduct experiments on the NLP task. Specifically, we fine-tune the GPT-2 model (Radford et al., 2018) on the E2E NLG Challenge for 10 epochs with batch size $B = 1000$, and report the standard metrics such as BLUE, ROUGE-L, etc., used in Hu et al. (2021) for evaluation. The results in Table 3 show that DiceSGD has better performance than DPSGD-GC.

To summarize the results of our experiments, we see that in both image classification and the NLP tasks, DiceSGD outperforms DPSGD-GC, and sometimes by a significant margin.

## 5 CONCLUSION

In this paper, we propose the DiceSGD algorithm for DP training. The algorithm uses a clipped error-feedback mechanism to eliminate the bias in gradient clipping. We provide novel convergence analysis in the strongly convex setting or under PL condition for DiceSGDwith a problem-independent clipping threshold and provide the DP guarantee independent of the problem type. Numerical results show superior performances of DiceSGD compared with DPSGD-GC on image classification and NLP tasks and the robustness of DiceSGD to the clipping threshold.

Table 3: Scores of fine-tuning GPT-2 on E2E NLG Challenge, with $C = 1.0$ and $(8, 8 \times 10^{-6})$-DP.

| Algorithm | BLEU | NIST | METEOR | ROUGE-L | CIDEr |
|---|---|---|---|---|---|
| DPSGD-GC | 56.8 | 4.83 | 36.2 | 65.2 | 1.43 |
| DiceSGD | 62.6 | 7.05 | 38.5 | 66.6 | 1.83 |
| SGD (Hu et al., 2021) | 70.4 | 8.85 | 46.8 | 71.8 | 2.53 |

ACKNOWLEDGEMENT

M. Hong and X. Zhang are supported by NSF grants CCF 1910385 and EPCN 2311007. X. Zhang is supported by Doctoral Dissertation Fellowship 2023, University of Minnesota.

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

| Symbol | Meaning |
|---|---|
| $\mathbf{x}, \mathbf{w}, \mathbf{e}$ | model variable, privacy noise, feedback error |
| $f(\cdot)$ | objective function |
| $\mathcal{D}, \mathcal{B}, \xi$ | dataset, minibatch, data sample |
| $N, B, i$ | Dataset size, minibatch size, data index |
| $\mathbf{g}_i, \nabla f(\mathbf{x}; \xi_i)$ | gradient evaluated on $\xi_i$ |
| $\epsilon, \delta$ | parameters of $(\epsilon, \delta)$-DP |
| $T, t$ | total iteration, iteration index |
| $\mu, L, \sigma$ | strong convexity (PL condition), Lipschitz, variance constants |
| $C, C_1, C_2$ | clipping thresholds |
| $\alpha_i^t, \alpha_{\mathbf{e}}^t$ | clipping factors of $\mathbf{g}_i^t, \mathbf{e}^t$, i.e., $\min\{1, \frac{C_1}{\|\mathbf{g}_i^t\|}\}, \min\{1, \frac{C_2}{\|\mathbf{e}^t\|}\}$ |
| $\mathcal{A}_1^t, \mathcal{A}_2^t$ | update of $\mathbf{x}^{t+1}$ and $\mathbf{e}^{t+1}$, i.e. lines 4,5,6 and 4,5,7 of Algorithm 2 |
| $\mathcal{H}^t$ | sequence of $\{\mathbf{x}^0, \mathbf{x}^1, \ldots, \mathbf{x}^t\}$ |
| $\Delta_{\mathbf{g}}^t$ | difference of $\mathbf{g}^t$ with $\mathcal{D}, \mathcal{D}'$, i.e., $\frac{1}{B}\sum_{i \in \mathcal{B}^t} \text{clip}(\mathbf{g}_i^t, C_1) - \frac{1}{B}\sum_{i \in \mathcal{B}'^t} \text{clip}(\mathbf{g}_i^t, C_1)$ |
| $\Delta_{\mathbf{e}}^t$ | update difference of $\mathbf{e}^t$ with neighboring datasets, i.e., $\mathbf{e}^t - \mathbf{e}'^t$. |

Table 4: Symbols used in the paper.

## A    PROOF OF RESULTS IN SECTION 3

In this section, we provide detailed proofs of the theorems in Section 3. We find the following relations useful:

$$\|a + b\|^2 \leq (1 + \beta)\|a\|^2 + \left(1 + \frac{1}{\beta}\right)\|b\|^2, \forall \beta > 0 \tag{5}$$

$$\langle a, b \rangle = \frac{1}{2}\left(\|a\|^2 + \|b\|^2 - \|a - b\|^2\right) \tag{6}$$

**Lemma A.1.** *Given a random variable $X \in \mathbb{R}^d$. If $c(\cdot) : \mathbb{R}^d \to \mathbb{R}^d$ is a non-expansive mapping, such that $\|c(a) - c(b)\| \leq \|a - b\|$, then $\text{Var}(c(X)) \leq \text{Var}(X)$. Additionally, function $c(x) = x - \text{clip}(x, C)$ is a non-expansive mapping for any positive constant clipping threshold $C > 0$.*

The proof is given in Appendix A.4.

**Theorem A.2.** *Assume the problem satisfies Assumption 3.1, 3.2, 3.4, and 3.5. Given any constant DP noise multiplier $\sigma_1$, by running DiceSGD (Algorithm 2) for $T$ iterations, choosing stepsize $\eta = \sqrt{\frac{2(f(\mathbf{x}^0) - f^\star)}{TL(2C_1^2 + 3C_2^2 + d\sigma_1^2)}}$, clipping thresholds $C_2 \geq 3C_1 + \frac{\sigma}{B} > 0$. It satisfies*

$$\mathbb{E}_t\left[\|\nabla f(\mathbf{x}^t)\|^2\right] \leq 2\sqrt{\frac{2L(f(\mathbf{x}^0) - f^\star)(2C_1^2 + 3C_2^2 + d\sigma_1^2)}{T}}, \tag{7}$$

*where the expectation $\mathbb{E}_t$ is taken over $t \in \{0, \ldots, T - 1\}$, with probability $A_t / \sum_{t=0}^{T-1} A_t$, with $\{A_t\} \in (0, 1]$ being a positive sequence defined in (12).*

**Theorem A.3.** *Under Assumptions 3.4 and 3.5, given constant $C$, choose the clipping thresholds $0 < C_1 \leq C_2 \leq C/B$. Choosing DP noise multiplier $\sigma_1$ as*

$$\sigma_1^2 \geq \frac{32T(C_1^2 + 2\min\{C^2, G'^2\})\log(1/\delta)}{N^2\epsilon^2}.$$

*Running DiceSGD for $T$ iteration, the algorithm guarantees $(\epsilon, \delta)$-DP.*

Let us denote the clipping factors as $\alpha_i^t := \min\left\{1, \frac{C_1}{\|\mathbf{g}_i^t\|}\right\}, \alpha_{\mathbf{e}}^t := \min\left\{1, \frac{C_2}{\|\mathbf{e}^t\|}\right\}$, so that

$$\text{clip}(\mathbf{e}^t, C_2) = \alpha_{\mathbf{e}}^t \mathbf{e}^t, \text{ clip}(\mathbf{g}_i^t, C_1) = \alpha_i^t \mathbf{g}_i^t, \text{ and } \mathbf{e}^{t+1} = (1 - \alpha_{\mathbf{e}}^t)\mathbf{e}^t + \frac{1}{B}\sum_{i \in \mathcal{B}^t}(1 - \alpha_i^t)\mathbf{g}_i^t. \tag{8}$$

Note that we have the following bound on $\alpha_{\mathbf{e}}^t$: Let us consider two cases: 1) $\|\mathbf{e}^t\| \leq C_2$ and $\|\mathbf{e}^t\| > C_2$. For case 1), it is clear that $\alpha_{\mathbf{e}}^t = 1$. For case 2), we have $\alpha_{\mathbf{e}}^t = \frac{C_2}{\|\mathbf{e}^t\|}$. We can further

bound $\|\mathbf{e}^t\|$ recursively as

$$\|\mathbf{e}^t\| \overset{(8)}{=} \left\| (1 - \alpha_{\mathbf{e}}^{t-1})\mathbf{e}^{t-1} + \frac{1}{B} \sum_{i \in \mathcal{B}^{t-1}} (1 - \alpha_i^{t-1}) g_i^{t-1} \right\|$$

$$\leq (1 - \alpha_{\mathbf{e}}^{t-1}) \left\| \mathbf{e}^{t-1} \right\| + \left\| \frac{1}{B} \sum_{i \in \mathcal{B}^{t-1}} (1 - \alpha_i^{t-1}) g_i^{t-1} \right\|$$

$$\overset{(a)}{\leq} (1 - \alpha_{\mathbf{e}}^{t-1}) \left\| \mathbf{e}^{t-1} \right\| + \frac{1}{B} \sum_{i \in \mathcal{B}^{t-1}} \left\| \left(1 - \min\{1, \frac{C_1}{\|g_i^{t-1}\|}\}\right) g_i^{t-1} \right\|$$

$$\overset{(b)}{\leq} (1 - \alpha_{\mathbf{e}}^{t-1}) \left\| \mathbf{e}^{t-1} \right\| + \frac{1}{B} \sum_{i \in \mathcal{B}^{t-1}} \max\{0, \|g_i^{t-1}\| - C_1\}$$

$$\overset{(c)}{\leq} \left\| \mathbf{e}^{t-1} \right\| - C_2 + \max\{0, \|\nabla f(\mathbf{x}^{t-1})\| + \sigma - C_1\}$$

$$\overset{(d)}{\leq} \left\| \mathbf{e}^1 \right\| + \sum_{\tau=1}^{t-1} (\max\{0, \|\nabla f(\mathbf{x}^\tau)\| + \sigma - C_1\} - C_2)$$

$$\overset{(e)}{\leq} t \max\{0, G + \sigma - C_1\} - (t-1)C_2 = tG' - (t-1)C_2.$$

where $(a)$ substitutes the definition of $\alpha_{\mathbf{e}}^t$; $(b)$ expands the last term; $(c)$ applies Assumption 3.4; in $(d)$ we recursively expand $\|\mathbf{e}^{t-1}\|$ to $\|\mathbf{e}^1\|$ and notice that $\|\mathbf{e}^1\| = \|\frac{1}{B} \sum_{i \in \mathcal{B}^1} (1 - \alpha_i^1) g_i^1\| \leq \max\{0, \|\nabla f(\mathbf{x}^1)\| + \sigma - C_1\}$; in $(e)$ we apply Assumption 3.5 and define $G' := \max\{0, G + \sigma - C_1\}$. Therefore, we have

$$\alpha_{\mathbf{e}}^t = \frac{C_2}{\|\mathbf{e}^t\|} \geq \frac{C_2}{tG' - (t-1)C_2} = \frac{C_2}{C_2 + t(G' - C_2)}. \tag{9}$$

## A.1 Proof of Theorem 3.6

With smoothness Assumption 3.2, we have the following descent property, where the expectation $\mathbb{E}_t[\cdot]$ is taken on the randomness over iteration $t$ conditioned on all past histories from $0$ to $t$:

$$\mathbb{E}_t[f(\mathbf{x}^{t+1})] \leq f(\mathbf{x}^t) + \langle \nabla f(\mathbf{x}^t), \mathbb{E}_t[\mathbf{x}^{t+1} - \mathbf{x}^t] \rangle + \mathbb{E}_t\left[ \frac{L}{2} \|\mathbf{x}^{t+1} - \mathbf{x}^t\|^2 \right]$$

$$\overset{(a)}{\leq} f(\mathbf{x}^t) - \eta \langle \nabla f(\mathbf{x}^t), \mathbb{E}_t[\mathbf{v}^t] \rangle + \frac{L\eta^2}{2} \mathbb{E}_t\left[ \|\mathbf{v}^t\|^2 \right] + \frac{Ld\eta^2}{2} \sigma_1^2$$

$$= f(\mathbf{x}^t) - \eta \langle \nabla f(\mathbf{x}^t), \mathbb{E}_t[\mathbf{v}^t] \rangle + \frac{L\eta^2}{2} \mathbb{E}_t\left[ \left\| \frac{1}{B} \sum_{i \in \mathcal{B}^t} \text{clip}\left(\mathbf{g}_i^t, C_1\right) + \text{clip}\left(\mathbf{e}^t, C_2\right) \right\|^2 \right] + \frac{Ld\eta^2}{2} \sigma_1^2$$

$$\overset{(b)}{\leq} f(\mathbf{x}^t) - \eta \langle \nabla f(\mathbf{x}^t), \mathbb{E}_t[\mathbf{v}^t] \rangle + \frac{L\eta^2}{2}(C_1 + C_2)^2 + \frac{Ld\eta^2}{2} \sigma_1^2, \tag{10}$$

where $(a)$ applies the update rule of $\mathbf{x}^t$ and uses the fact that $\mathbf{w}^t$ follows zero-mean Gaussian and is independent of $\mathbf{v}^t$, and $(b)$ bounds $\|\mathbf{v}^t\|$ by $C_1 + C_2$ with clipping operation. Next, we bound the inner-product between $\nabla f(\mathbf{x}^t)$ and $\mathbb{E}[\mathbf{v}^t]$ as:

$$-\langle \nabla f(\mathbf{x}^t), \mathbb{E}[\mathbf{v}^t] \rangle \overset{(8)}{=} -\left\langle \nabla f(\mathbf{x}^t), \mathbb{E}_t \frac{1}{B} \sum_{i \in \mathcal{B}^t} \alpha_i^t \mathbf{g}_i^t + \mathbb{E}_t \alpha_{\mathbf{e}}^t \mathbf{e}^t \right\rangle =$$

$$= -\langle \nabla f(\mathbf{x}^t), \mathbb{E} \alpha_i^t \mathbf{g}_i^t + \mathbb{E} \alpha_{\mathbf{e}}^t \mathbf{e}^t \rangle$$

$$= -\langle \nabla f(\mathbf{x}^t), \mathbb{E} \alpha_i^t \mathbf{g}_i^t \rangle - \langle \nabla f(\mathbf{x}^t) - \nabla f(\mathbf{x}^{t-1}), \mathbb{E} \alpha_{\mathbf{e}}^t \mathbf{e}^t \rangle - \langle \nabla f(\mathbf{x}^{t-1}), \mathbb{E} \alpha_{\mathbf{e}}^t \mathbf{e}^t \rangle$$

$$\overset{(a)}{\leq} -\langle \nabla f(\mathbf{x}^t), \mathbb{E} \alpha_i^t \mathbf{g}_i^t \rangle + \frac{1}{2}\left( \frac{1}{\eta L} \mathbb{E} \|\nabla f(\mathbf{x}^t) - \nabla f(\mathbf{x}^{t-1})\|^2 + \eta L \|\alpha_{\mathbf{e}}^t \mathbf{e}^t\|^2 \right)$$

$$- \left\langle \nabla f(\mathbf{x}^{t-1}), \mathbb{E} \alpha_{\mathbf{e}}^t \left( (1 - \alpha_{\mathbf{e}}^{t-1})\mathbf{e}^{t-1} + \nabla f(\mathbf{x}^{t-1}) - \frac{1}{N} \sum_{i=1}^N \alpha_i^{t-1} \mathbf{g}_i^{t-1} \right) \right\rangle$$

$$\overset{(b)}{\leq} -\left\langle \nabla f(\mathbf{x}^t), \mathbb{E}\,\alpha_i^t \mathbf{g}_i^t \right\rangle + \frac{\eta L}{2}\left( \mathbb{E}\left\| \mathbf{v}^{t-1} + \mathbf{w}^{t-1} \right\|^2 + \mathbb{E}\left\| \alpha_{\mathbf{e}}^t \mathbf{e}^t \right\|^2 \right)$$

$$- \frac{\alpha_{\mathbf{e}}^t(1 - \alpha_{\mathbf{e}}^{t-1})}{\alpha_{\mathbf{e}}^{t-1}} \left\langle \nabla f(\mathbf{x}^{t-1}), \mathbb{E}\,\alpha_{\mathbf{e}}^{t-1} \mathbf{e}^{t-1} \right\rangle - \alpha_{\mathbf{e}}^t \left\| \nabla f(\mathbf{x}^{t-1}) \right\|^2 + \alpha_{\mathbf{e}}^t \left\langle \nabla f(\mathbf{x}^{t-1}), \mathbb{E}\,\alpha_i^{t-1} \mathbf{g}_i^{t-1} \right\rangle$$

$$\overset{(c)}{\leq} -\left\langle \nabla f(\mathbf{x}^t), \mathbb{E}\,\alpha_i^t \mathbf{g}_i^t \right\rangle + \frac{\eta L}{2}\left( (C_1 + C_2)^2 + d\sigma_1^2 + C_2^2 \right) - \alpha_{\mathbf{e}}^t \left\| \nabla f(\mathbf{x}^{t-1}) \right\|^2$$

$$- \frac{\alpha_{\mathbf{e}}^t(1 - \alpha_{\mathbf{e}}^{t-1})}{\alpha_{\mathbf{e}}^{t-1}} \left\langle \nabla f(\mathbf{x}^{t-1}), \mathbb{E}\,\alpha_{\mathbf{e}}^{t-1} \mathbf{e}^{t-1} \right\rangle + \alpha_{\mathbf{e}}^t \left\langle \nabla f(\mathbf{x}^{t-1}), \mathbb{E}\,\alpha_i^{t-1} \mathbf{g}_i^{t-1} \right\rangle,$$

where $(a)$ applies (6) to the second term and expand $\mathbf{e}^t$ with (8); $(b)$ applies the smoothness Assumption 3.2 to the second term; $(c)$ uses the fact that $\mathbf{w}^{t-1}$ follows zero-mean Gaussian and is independent of $\mathbf{v}^{t-1}$ and that $\mathbf{v}^{t-1} = \mathrm{clip}\left(\mathbf{e}^{t-1}, C_2\right) + \frac{1}{B}\sum_{i \in \mathcal{B}^{t-1}} \mathrm{clip}\left(\mathbf{g}_i^{t-1}, C_1\right)$ has magnitude less or equal to $C_1 + C_2$.

By recursively expanding the term $-\left\langle \nabla f(\mathbf{x}^\tau), \mathbb{E}\,\alpha_{\mathbf{e}}^\tau \mathbf{e}^\tau \right\rangle$ above as

$$-\left\langle \nabla f(\mathbf{x}^\tau), \mathbb{E}\,\alpha_{\mathbf{e}}^\tau \mathbf{e}^\tau \right\rangle \leq \frac{\eta L}{2}\left( (C_1 + C_2)^2 + d\sigma_1^2 + C_2^2 \right) - \alpha_{\mathbf{e}}^\tau \left\| \nabla f(\mathbf{x}^{\tau-1}) \right\|^2$$

$$- \frac{\alpha_{\mathbf{e}}^t(1 - \alpha_{\mathbf{e}}^{\tau-1})}{\alpha_{\mathbf{e}}^{\tau-1}} \left\langle \nabla f(\mathbf{x}^{\tau-1}), \mathbb{E}\,\alpha_{\mathbf{e}}^{\tau-1} \mathbf{e}^{\tau-1} \right\rangle + \alpha_{\mathbf{e}}^\tau \left\langle \nabla f(\mathbf{x}^{\tau-1}), \mathbb{E}_{\tau-1}\,\alpha_i^{\tau-1} \mathbf{g}_i^{\tau-1} \right\rangle,$$

and note that $\mathbf{e}^0 = 0, \alpha_{\mathbf{e}}^0 = 1$, we have:

$$-\left\langle \nabla f(\mathbf{x}^t), \mathbb{E}[\mathbf{v}^t] \right\rangle = -\left\langle \nabla f(\mathbf{x}^t), \mathbb{E}\,\alpha_i^t \mathbf{g}_i^t \right\rangle - \left\langle \nabla f(\mathbf{x}^t), \mathbb{E}\,\alpha_{\mathbf{e}}^t \mathbf{e}^t \right\rangle$$

$$\leq -\left\langle \nabla f(\mathbf{x}^t), \mathbb{E}\,\alpha_i^t \mathbf{g}_i^t \right\rangle + \frac{\eta L}{2}\left( (C_1 + C_2)^2 + d\sigma_1^2 + C_2^2 \right) - \alpha_{\mathbf{e}}^t \left\| \nabla f(\mathbf{x}^{t-1}) \right\|^2$$

$$- \frac{\alpha_{\mathbf{e}}^t(1 - \alpha_{\mathbf{e}}^{t-1})}{\alpha_{\mathbf{e}}^{t-1}} \left\langle \nabla f(\mathbf{x}^{t-1}), \mathbb{E}\,\alpha_{\mathbf{e}}^{t-1} \mathbf{e}^{t-1} \right\rangle + \alpha_{\mathbf{e}}^t \left\langle \nabla f(\mathbf{x}^{t-1}), \mathbb{E}\,\alpha_i^{t-1} \mathbf{g}_i^{t-1} \right\rangle$$

$$\leq -\sum_{\tau=0}^{t-1} \alpha_{\mathbf{e}}^t \left( \prod_{\tau_1=\tau+1}^{t-1} (1 - \alpha_{\mathbf{e}}^{\tau_1}) \right) \left\| \nabla f(\mathbf{x}^\tau) \right\|^2 - \left\langle \nabla f(\mathbf{x}^t), \mathbb{E}\,\alpha_i^t \mathbf{g}_i^t \right\rangle$$

$$+ \sum_{\tau=1}^{t} \alpha_{\mathbf{e}}^t \left( \prod_{\tau_1=\tau}^{t-1} (1 - \alpha_{\mathbf{e}}^{\tau_1}) \right) \left( \frac{L\eta}{2}\left( 2C_1^2 + 3C_2^2 + d\sigma_1^2 \right) + \left\langle \nabla f(\mathbf{x}^\tau), \mathbb{E}\,\alpha_i^\tau \mathbf{g}_i^\tau \right\rangle \right). \quad (11)$$

Substitute (11) back to (10) and sum over iterations, we have:

$$\mathbb{E}[f(\mathbf{x}^T)] \leq f(\mathbf{x}^0) - \eta \sum_{t=0}^{T-1} \sum_{\tau=0}^{t-1} \alpha_{\mathbf{e}}^t \left( \prod_{\tau_1=\tau+1}^{t-1} (1 - \alpha_{\mathbf{e}}^{\tau_1}) \right) \left\| \nabla f(\mathbf{x}^\tau) \right\|^2$$

$$+ \frac{L\eta^2}{2} \sum_{t=0}^{T-1} \left( \sum_{\tau=1}^{t} \alpha_{\mathbf{e}}^t \left( \prod_{\tau_1=\tau}^{t-1} (1 - \alpha_{\mathbf{e}}^{\tau_1}) \right) \left( 2C_1^2 + 3C_2^2 + d\sigma_1^2 \right) + 2C_1^2 + 2C_2^2 + d\sigma_1^2 \right)$$

$$- \eta \sum_{t=0}^{T-1} \left( \left\langle \nabla f(\mathbf{x}^t), \mathbb{E}\,\alpha_i^t \mathbf{g}_i^t \right\rangle - \sum_{\tau=1}^{t} \alpha_{\mathbf{e}}^t \left( \prod_{\tau_1=\tau}^{t-1} (1 - \alpha_{\mathbf{e}}^{\tau_1}) \right) \left\langle \nabla f(\mathbf{x}^\tau), \mathbb{E}\,\alpha_i^\tau \mathbf{g}_i^\tau \right\rangle \right)$$

$$\overset{(a)}{=} f(\mathbf{x}^0) - \eta \sum_{t=0}^{T-1} \sum_{\tau=t+1}^{T-1} \alpha_{\mathbf{e}}^\tau \left( \prod_{\tau_1=t+1}^{\tau-1} (1 - \alpha_{\mathbf{e}}^{\tau_1}) \right) \left\| \nabla f(\mathbf{x}^t) \right\|^2$$

$$+ \frac{L\eta^2}{2} \sum_{t=0}^{T-1} \left( \sum_{\tau=1}^{t} \alpha_{\mathbf{e}}^t \left( \prod_{\tau_1=\tau}^{t-1} (1 - \alpha_{\mathbf{e}}^{\tau_1}) \right) \left( 2C_1^2 + 3C_2^2 + d\sigma_1^2 \right) + 2C_1^2 + 2C_2^2 + d\sigma_1^2 \right)$$

$$- \eta \sum_{t=0}^{T-1} \left( 1 - \sum_{\tau=t+1}^{T-1} \alpha_{\mathbf{e}}^\tau \left( \prod_{\tau_1=t+1}^{\tau-1} (1 - \alpha_{\mathbf{e}}^{\tau_1}) \right) \right) \left\langle \nabla f(\mathbf{x}^t), \mathbb{E}\,\alpha_i^t \mathbf{g}_i^t \right\rangle$$

$$\overset{(b)}{\leq} f(\mathbf{x}^0) - \eta \sum_{t=0}^{T-1} A_t \left\| \nabla f(\mathbf{x}^t) \right\|^2 + \frac{L\eta^2}{2} B_T \left( 2C_1^2 + 3C_2^2 + d\sigma_1^2 \right) + \eta \sum_{t=0}^{T-1} C_t C_1 \left\| \nabla f(\mathbf{x}^t) \right\|,$$

where in $(a)$ we rearrange the terms; in $(b)$ we apply Cauchy-Schwarz inequality to the last term and define $A_t, B_T, C_t$ accordingly. Specifically, we have

$$A_t := \sum_{\tau=t+1}^{T-1} \alpha_{\mathbf{e}}^\tau \left( \prod_{\tau_1=t+1}^{\tau-1} (1 - \alpha_{\mathbf{e}}^{\tau_1}) \right), \tag{12}$$

$$B_T := T + \sum_{t=0}^{T-1} \sum_{\tau=1}^{t} \alpha_{\mathbf{e}}^t \left( \prod_{\tau_1=\tau}^{t-1} (1 - \alpha_{\mathbf{e}}^{\tau_1}) \right) = T + \sum_{t=0}^{T-1} A_t, \tag{13}$$

$$C_t := \left| 1 - \sum_{\tau=t+1}^{T-1} \alpha_{\mathbf{e}}^\tau \left( \prod_{\tau_1=t+1}^{\tau-1} (1 - \alpha_{\mathbf{e}}^{\tau_1}) \right) \right| = |1 - A_t|. \tag{14}$$

Next, we bound $A_t, B_T, C_t$, respectively. To begin with, we provide the upper and lower bounds on $A_t$. First, note that by definition of $\alpha_{\mathbf{e}}^t$ in (8), we have $\alpha_{\mathbf{e}}^t \in (0, 1]$, therefore the product $\prod_{\tau_1=t+1}^{\tau-1} (1 - \alpha_{\mathbf{e}}^{\tau_1})$ is strictly less than 1. Let $T' \in \{0, \dots T - 1\}$ denote the iteration where $\alpha_{\mathbf{e}}^t < 1, \forall t > T'$ and $\alpha_{\mathbf{e}}^{T'} = 1$. Then $A_t$ can be expressed as:

$$A_t = \sum_{\tau=t+1}^{T-1} \alpha_{\mathbf{e}}^\tau \left( \prod_{\tau_1=t+1}^{\tau-1} (1 - \alpha_{\mathbf{e}}^{\tau_1}) \right)$$

$$= 1 - 1 + \alpha_{\mathbf{e}}^{t+1} + \alpha_{\mathbf{e}}^{t+2}(1 - \alpha_{\mathbf{e}}^{t+1}) + \cdots + \alpha_{\mathbf{e}}^{T-1} \prod_{\tau=t+1}^{T-2} (1 - \alpha_{\mathbf{e}}^\tau)$$

$$= 1 - (1 - \alpha_{\mathbf{e}}^{t+1}) + \alpha_{\mathbf{e}}^{t+2}(1 - \alpha_{\mathbf{e}}^{t+1}) + \cdots + \alpha_{\mathbf{e}}^{T-1} \prod_{\tau=t+1}^{T-2} (1 - \alpha_{\mathbf{e}}^\tau)$$

$$= 1 - \prod_{\tau=t+1}^{T-1} (1 - \alpha_{\mathbf{e}}^\tau) = \begin{cases} 1, & t < T', \\ 1 - \prod_{\tau=t+1}^{T-1} (1 - \alpha_{\mathbf{e}}^\tau), & t \geq T'. \end{cases}$$

Therefore, $B_T \leq 2T$. $C_t = 1 - A_t \in [0, 1)$. Next, we show that the following relation holds: Case I: $t < T', A_t = 1$, it is clear that

$$A_t \left\| \nabla f(\mathbf{x}^t) \right\|^2 \geq (1 - A_t) C_1 \left\| \nabla f(\mathbf{x}^t) \right\| = 0.$$

Case II: $t \geq T'$, then $A_t < 1$, we want to show the following relation holds:

$$\alpha_{\mathbf{e}}^{t+1} \sum_{\tau=T'}^{t} \prod_{\tau_1=\tau+1}^{t} (1 - \alpha_{\mathbf{e}}^{\tau_1}) \left\| \nabla f(\mathbf{x}^\tau) \right\|^2 > (1 - A_t) C_1 \left\| \nabla f(\mathbf{x}^t) \right\|.$$

Let us consider the worst case where $t = T'$, i.e., when the left-hand-side only has one term $\alpha_{\mathbf{e}}^{t+1} \left\| \nabla f(\mathbf{x}^t) \right\|^2$, then, we have

$$\alpha_{\mathbf{e}}^{t+1} = \frac{C_2}{\|\mathbf{e}^{t+1}\|} < 1$$

$$\frac{C_2}{\left\| \mathbf{e}^t - \alpha_{\mathbf{e}}^t \mathbf{e}^t + \frac{1}{B} \sum_{i \in \mathcal{B}^t} (\mathbf{g}_i^t - \alpha_i^t \mathbf{g}_i^t) \right\|} \overset{(8)}{<} 1$$

$$C_2 \overset{(a)}{<} \left\| \mathbf{e}^t - 1 \cdot \mathbf{e}^t + \frac{1}{B} \sum_{i \in \mathcal{B}^t} (\mathbf{g}_i^t - \alpha_i^t \mathbf{g}_i^t) \right\|$$

$$C_2 \overset{(b)}{<} \left\| \nabla f(\mathbf{x}^t) + \frac{1}{B} \sum_{i \in \mathcal{B}^t} (\mathbf{g}_i^t - \nabla f(\mathbf{x}^t) - \alpha_i^t \mathbf{g}_i^t) \right\|$$

$$C_2 \overset{(c)}{<} \left\| \nabla f(\mathbf{x}^t) \right\| + \frac{\sigma}{B} + \frac{1}{B} \sum_{i \in \mathcal{B}^t} \|\alpha_i^t \mathbf{g}_i^t\|$$

$$C_2 - \frac{\sigma}{B} - C_1 \overset{(d)}{<} \left\| \nabla f(\mathbf{x}^t) \right\|, \tag{15}$$

where $(a)$ uses the fact that $\alpha_{\mathbf{e}}^t = 1$ at $t = T'$; $(b)$ we add and subtract $\nabla f(\mathbf{x}^t)$; $(c)$ applies triangle inequality and Assumption 3.4; and $(d)$ we arrange the terms and use the fact that $\|\alpha_i^t \mathbf{g}_i^t\| \leq C_1$. By setting $C_2 \geq 3C_1 + \frac{\sigma}{B}$, we have $\|\nabla f(\mathbf{x}^t)\| \geq 2C_1$. Further, from (15), we also have that

$$
\begin{aligned}
\alpha_{\mathbf{e}}^{t+1} &= \frac{C_2}{\left\|\mathbf{e}^t - \alpha_{\mathbf{e}}^t \mathbf{e}^t + \frac{1}{B}\sum_{i\in\mathcal{B}^t}(\mathbf{g}_i^t - \alpha_i^t \mathbf{g}_i^t)\right\|} \\
&\geq \frac{C_2}{\|\nabla f(\mathbf{x}^t)\| + \frac{\sigma}{B} + C_1},
\end{aligned}
\tag{16}
$$

where we apply triangle inequality to the denominator in the second inequality. Therefore, we have for the worst case:

$$
\begin{aligned}
\alpha_{\mathbf{e}}^{t+1}\left\|\nabla f(\mathbf{x}^t)\right\|^2 &\overset{(16)}{\geq} \frac{C_2}{\|\nabla f(\mathbf{x}^t)\| + C_1 + \frac{\sigma}{B}}\left\|\nabla f(\mathbf{x}^t)\right\| \cdot \left\|\nabla f(\mathbf{x}^t)\right\| \\
&\overset{(a)}{\geq} \frac{(3C_1 + \frac{\sigma}{B})2C_1}{3C_1 + \frac{\sigma}{B}}\left\|\nabla f(\mathbf{x}^t)\right\| \geq 2C_1\left\|\nabla f(\mathbf{x}^t)\right\| > 2(1-A_t)C_1\left\|\nabla f(\mathbf{x}^t)\right\|,
\end{aligned}
$$

where $(a)$ uses the fact that $\frac{C_2}{\|\nabla f(\mathbf{x}^t)\| + C_1 + \frac{\sigma}{B}}\|\nabla f(\mathbf{x}^t)\|$ is monotonically increasing w.r.t. $\|\nabla f(\mathbf{x}^t)\|$. When $t > T'$, a similar proof can be applied to show that $\alpha_{\mathbf{e}}^{t+1}\sum_{\tau=T'}^t \prod_{\tau_1=\tau+1}^t (1 - \alpha_{\mathbf{e}}^{\tau_1})\|\nabla f(\mathbf{x}^\tau)\| \geq 2C_1$. Putting the above results together, we have:

$$
\begin{aligned}
\mathbb{E}[f(\mathbf{x}^T)] &\leq f(\mathbf{x}^0) - \frac{\eta}{2}\sum_{t=0}^{T-1} A_t\left\|\nabla f(\mathbf{x}^t)\right\|^2 + \frac{TL\eta^2}{2}(2C_1^2 + 3C_2^2 + d\sigma_1^2), \\
\mathbb{E}_t[\|\nabla f(\mathbf{x}^t)\|^2] &\leq \frac{2(f(\mathbf{x}^0) - f^\star)}{\eta T} + \eta L(2C_1^2 + 3C_2^2 + d\sigma_1^2),
\end{aligned}
\tag{17}
$$

where $C_2 \geq 3C_1 + \frac{\sigma}{B}$, and the expectation is taken over $t \in \{0,\ldots,T-1\}$, with probability $A_t / \sum_{t=0}^{T-1} A_t$. The theorem is proved.

## A.2  PROOF OF THEOREM 3.7

In this section, we provide the privacy analysis for DiceSGD algorithm and the proof of Theorem 3.7. Specifically, we adopt the Rényi differential privacy (RDP) notion for our analysis. The definition of Rényi-DP is given as follows:

**Definition A.4** (Rényi-DP Mironov (2017))**.** A randomized mechanism $\mathcal{M}$ is said to guarantee $(\alpha, \epsilon)$-RDP with order $\alpha > 1$, if for any two neighboring datasets $\mathcal{D}, \mathcal{D}'$ ($\mathcal{D}, \mathcal{D}'$ differ by one sample instance), it holds that

$$
D_\alpha(\mathcal{M}(\mathcal{D})\|\mathcal{M}(\mathcal{D}')) = \frac{1}{\alpha - 1}\log\mathbb{E}_{\theta\sim\mathcal{M}(\mathcal{D}')}\left[\left(\frac{\mathcal{M}(\mathcal{D})(\theta)}{\mathcal{M}(\mathcal{D}')(\theta)}\right)^\alpha\right] \leq \epsilon.
$$

Rényi-DP can be translated into the more popular $(\epsilon, \delta)$-DP Def. 2.1 with the following lemma:

**Lemma A.5** (Proposition 3 Mironov (2017) )**.** *A randomized mechanism $\mathcal{M}$ guarantees $(\alpha, \epsilon)$-RDP, then it guarantees $(\epsilon + \log(1/\delta)/(\alpha - 1), \delta)$-DP for all $\delta \in (0,1)$.*

To derive a privacy guarantee for the proposed DiceSGD algorithm, we first reformulate the algorithm as the following procedures. In specific, we define

$$
\mathcal{A}_1^t : \mathcal{X} \times \mathcal{E}^t \times \mathcal{D} \to \mathcal{X} \text{ as the evolution of } \mathbf{x}^t, \text{ i.e., lines 4,5,6 of Algorithm 2}
$$

$$
\mathcal{A}_2^t : \mathcal{X} \times \mathcal{E}^t \times \mathcal{D} \to \mathcal{E}^{t+1} \text{ as the evolution of } \mathbf{e}^t, \text{ i.e., lines 4,5,7 of Algorithm 2}
$$

$$
\mathcal{H}^t : \mathcal{D} \to \prod_{\tau=1}^t \mathcal{X} \text{ as the sequential observation of } \mathbf{x}^t, \text{ i.e., } \{\mathbf{x}^0, \mathbf{x}^1, \ldots, \mathbf{x}^t\}.
$$

In addition, we define $p = \frac{B}{N}$ and $\mathcal{D}, \mathcal{D}'$ be the two neighboring datasets where $\mathcal{D}'$ contains a unique sample $\xi'$, i.e., $\mathcal{D}' = \mathcal{D} \cup \{\xi'\}$. Then we recursively bound the RDP guarantee of $\mathcal{H}^t$ for $t = 1, \ldots, T$ with three steps.

We first provide the sketch of the proof as follows:

1. We show that the output sequence at the first iteration $\mathcal{H}^1$ satisfies $(\alpha, \epsilon^1)$-RDP by using the RDP of the sub-sampled Gaussian mechanism.

2. We assume that the output of $\mathcal{A}_1^t$ conditioned on the past output sequence $\mathcal{H}^t$ satisfies $(\alpha, \epsilon')$-RDP. Then we can derive the RDP guarantee for the output sequence at iteration $t + 1$, i.e., $\mathcal{H}^{t+1} = \{\mathcal{H}^t, \mathcal{A}_1^t | \mathcal{H}^t\}$ by the composition theorem of RDP.

3. We provide the RDP bound for $\mathcal{A}_1^t$ conditioned on $\mathcal{H}^t$, which consists of the sub-sampled gradients at iteration $t$ and the update of $\mathbf{e}^t$ with $\mathcal{A}_2^\tau$ from iteration $\tau = 1, \ldots, t - 1$ combined with the Gaussian noise. Therefore, we bound the sensitivity and Rényi divergence of $\mathcal{A}_1^t$ by accounting for the impact of the neighboring datasets on both the sub-sampled gradients at iteration $t$ and the updates of $\mathbf{e}^t$ conditioned on $\mathcal{H}^t$. Using the update of $\mathbf{e}^t$, we can recursively derive the Rényi divergence of $\mathbf{e}^t$ from $\mathbf{e}^{t-1}, \ldots, \mathbf{e}^0$.

The above three steps enable us to bound the $(\alpha, \epsilon^t)$-RDP for $\mathcal{H}^t$, the output of DiceSGD algorithm. Finally, by applying Lemma A.5, we obtain the $(\epsilon, \delta)$-DP guarantee for DiceSGD algorithm.

**Step I:** First, when $t = 1$, $\mathcal{H}^1 = \mathcal{A}_1^1$. Apply Lemma A.10, and we obtain that $\mathcal{H}^1$ satisfies $(\alpha, \epsilon(\sigma))$-RDP where $\epsilon(\sigma) \leq \frac{8 C_1^2 \alpha}{N^2 \sigma_1^2}$ is a function depending on the size of the injected noise $\sigma$, and we assume $p \leq \frac{1}{5}$.

**Step II: Claim:** Suppose $\mathcal{H}^t$ satisfies $(\alpha, \epsilon)$-RDP, $\mathcal{A}_1^t$ conditioned on $\mathcal{H}^t$ satisfies $(\alpha, \epsilon')$-RDP, then $\mathcal{H}^{t+1}$ satisfies $(\alpha, \epsilon + \epsilon')$-RDP.

*Proof.* Let us define $X_{t+1}(\mathbf{x}^{t+1} | \mathcal{H}^t)$ and $X'_{t+1}(\mathbf{x}^{t+1} | \mathcal{H}^t)$ be the conditional probability-density-function (PDF) of the output of $\mathcal{A}_1^t$ with neighboring datasets $\mathcal{D}$ and $\mathcal{D}'$ conditioned on the past outputs, respectively; similarly, define $H_t(\mathcal{H}^t), H'_t(\mathcal{H}^t)$ be the PDF of the output of $\mathcal{H}^t$ with datasets $\mathcal{D}$ and $\mathcal{D}'$. Then $H_{t+1}(\mathcal{H}^{t+1}) = H_{t+1}(\mathcal{H}^t, \mathbf{x}^{t+1}) = H_t(\mathcal{H}^t) X_{t+1}(\mathbf{x}^{t+1} | \mathcal{H}^t)$, and we have

$$\exp[(\alpha - 1) D_\alpha(\mathcal{H}^{t+1}(D) \| \mathcal{H}^{t+1}(D'))]$$

$$= \int_{\prod_0^t \mathcal{X}} \int_{\mathcal{X}} H_t(\mathcal{H}^t)^\alpha H'_t(\mathcal{H}^t)^{1-\alpha} X_{t+1}(\mathbf{x}^{t+1} | \mathcal{H}^t)^\alpha X'_{t+1}(\mathbf{x}^{t+1} | \mathcal{H}^t)^{1-\alpha} \mathrm{d}\mathcal{H} \mathrm{d}\mathbf{x}^{t+1}$$

$$= \int_{\prod_0^t \mathcal{X}} H_t(\mathcal{H}^t)^\alpha H'_t(\mathcal{H}^t)^{1-\alpha} \int_{\mathcal{X}} X_{t+1}(\mathbf{x}^{t+1} | \mathcal{H}^t)^\alpha X'_{t+1}(\mathbf{x}^{t+1} | \mathcal{H}^t)^{1-\alpha} \mathrm{d}\mathbf{x}^{t+1} \mathrm{d}\mathcal{H}^t \quad (18)$$

$$\overset{(a)}{\leq} \exp((\alpha - 1)\epsilon) \exp((\alpha - 1)\epsilon')$$

$$= \exp((\alpha - 1)(\epsilon + \epsilon')),$$

where $(a)$ applies the assumptions that $\mathcal{H}^t$ satisfies $(\alpha, \epsilon)$-RDP and $\mathcal{X}^t$ satisfies $(\alpha, \epsilon')$-RDP to the first and the second integration, respectively. Thus $\mathcal{H}^{t+1}$ satisfies $(\alpha, \epsilon + \epsilon')$-RDP. □

**Step III:** In the above step, we use the assumption that conditioning on $\mathcal{H}^t$, $\mathcal{X}^t$ satisfies $(\alpha, \epsilon')$-RDP. In this step, we explicitly bound $\epsilon'$ in the $(\alpha, \epsilon')$-RDP of $\mathcal{A}_1^t$ conditioning on $\mathcal{H}^t$. We first expand the Rényi divergence of $\mathcal{A}_1(\mathbf{x}^t, \mathcal{D})$ and $\mathcal{A}_1(\mathbf{x}^t, \mathcal{D}')$ as

$$D_\alpha(\mathcal{A}_1(\mathbf{x}^t, \mathcal{D}) \| \mathcal{A}_1(\mathbf{x}^t, \mathcal{D}') | \mathcal{H}^t) = D_\alpha(\mathcal{N}(\mathbf{x}^t - \eta \mathbf{v}^t, \eta^2 \sigma_1^2 \cdot I) \| \mathcal{N}(\mathbf{x}^t - \eta \mathbf{v}'^t, \eta^2 \sigma_1^2 \cdot I) | \mathcal{H}^t)$$

$$\overset{(a)}{=} D_\alpha(\mathcal{N}(\mathbf{v}^t - \mathbf{v}'^t, \sigma_1^2 \cdot I) \| \mathcal{N}(0, \sigma_1^2 \cdot I) | \mathcal{H}^t), \quad (19)$$

where in $(a)$ we first shift the mean of the Gaussian distributions by $-\mathbf{x}^t + \eta \mathbf{v}'^t$ and rescale them by a factor of $-\frac{1}{\eta}$. Let $\mu_0$ be the PDF of $\mathcal{N}(0, \sigma_1^2 \cdot I)$ and $\mathcal{N}_\sigma(\cdot)$ denote $\mathcal{N}(\cdot, \sigma_1^2 \cdot I)$. Define

$$\Delta_{\mathbf{g}}^t := \frac{1}{B} \sum_{i \in \mathcal{B}^t} \mathrm{clip}\left(\mathbf{g}_i^t, C_1\right) - \frac{1}{B} \sum_{i \in \mathcal{B}'^t} \mathrm{clip}\left(\mathbf{g}_i^t, C_1\right)$$

$$\Delta_{\mathbf{e}}^t := \mathbf{e}^t - \mathbf{e}'^t.$$

Then $\mathbf{v} - \mathbf{v}'$ can be expressed as

$$\mathbf{v}^t - \mathbf{v}'^t = \mathrm{clip}\left(\mathbf{e}^t, C_2\right) - \mathrm{clip}\left(\mathbf{e}'^t, C_2\right) + \Delta_{\mathbf{g}}^t.$$

Substitute the above relation of $\mathbf{v} - \mathbf{v}'$ to (19), we have

$$
\begin{aligned}
D_\alpha(\mathcal{N}_\sigma(\mathbf{v}^t - \mathbf{v}'^t)\|\mu_0|\mathcal{H}^t) &= D_\alpha(\mathcal{N}_\sigma(\mathrm{clip}\left(\mathbf{e}^t, C_2\right) - \mathrm{clip}\left(\mathbf{e}'^t, C_2\right) + \Delta_{\mathbf{g}}^t)\|\mu_0|\mathcal{H}^t) \\
&\overset{(a)}{\leq} 2D_\alpha(\mathcal{N}_\sigma(\mathrm{clip}\left(\mathbf{e}^t, C_2\right) - \mathrm{clip}\left(\mathbf{e}'^t, C_2\right))\|\mu_0|\mathcal{H}^t) \\
&\quad + 2D_\alpha(\mathcal{N}_\sigma(\mathrm{clip}\left(\mathbf{e}^t, C_2\right) - \mathrm{clip}\left(\mathbf{e}'^t, C_2\right) + \Delta_{\mathbf{g}}^t)\|\mathcal{N}_\sigma(\mathrm{clip}\left(\mathbf{e}^t, C_2\right) - \mathrm{clip}\left(\mathbf{e}'^t, C_2\right))|\mathcal{H}^t) \\
&\overset{(b)}{=} 2D_\alpha(\mathcal{N}_\sigma(\mathrm{clip}\left(\mathbf{e}^t, C_2\right) - \mathrm{clip}\left(\mathbf{e}'^t, C_2\right))\|\mu_0|\mathcal{H}^t) + 2D_\alpha(\mathcal{N}_\sigma(\Delta_{\mathbf{g}}^t)\|\mu_0|\mathcal{H}^t),
\end{aligned}
\tag{20}
$$

where $(a)$ applies Lemma A.8 with $a = \mathrm{clip}\left(\mathbf{e}^t, C_2\right) - \mathrm{clip}\left(\mathbf{e}'^t, C_2\right) + \Delta_{\mathbf{g}}^t, b = 0, c = \mathrm{clip}\left(\mathbf{e}^t, C_2\right) - \mathrm{clip}\left(\mathbf{e}'^t, C_2\right)$; $(b)$ shifts the mean of the Gaussian distributions in the second term by $-\mathrm{clip}\left(\mathbf{e}^t, C_2\right) - \mathrm{clip}\left(\mathbf{e}'^t, C_2\right)$. Next, we bound the two terms in (20) separately.

**Bounding the first term in** (20)**:** To bound $D_\alpha(\mathcal{N}_\sigma(\mathrm{clip}\left(\mathbf{e}^t, C_2\right) - \mathrm{clip}\left(\mathbf{e}'^t, C_2\right))\|\mu_0|\mathcal{H}^t)$. First notice that clipping operation is non-expansive with factor $\alpha_{\mathbf{e}}^t$, so we have

$$
\begin{aligned}
&D_\alpha(\mathcal{N}_\sigma(\mathrm{clip}\left(\mathbf{e}^t, C_2\right) - \mathrm{clip}\left(\mathbf{e}'^t, C_2\right))\|\mu_0|\mathcal{H}^t) \\
&\leq (\alpha_{\mathbf{e}}^t)^2 D_\alpha(\mathcal{N}_\sigma(\mathbf{e}^t - \mathbf{e}'^t)\|\mu_0|\mathcal{H}^t) = (\alpha_{\mathbf{e}}^t)^2 D_\alpha(\mathcal{N}_\sigma(\Delta_{\mathbf{e}}^t)\|\mu_0|\mathcal{H}^t)
\end{aligned}
$$

Then, we start with bounding the update of $\mathbf{e}$ with the following lemma:

**Lemma A.6.** *Let $\mathbf{x}^t, \mathbf{e}^t, \mathbf{e}'^t$ be the input of $\mathcal{A}_2$. Then the Rényi divergence $D_\alpha(\mathcal{N}_\sigma(\Delta_{\mathbf{e}}^{t+1})\|\mu_0)$ can be bounded by*

$$
D_\alpha(\mathcal{N}_\sigma(\Delta_{\mathbf{e}}^{t+1})\|\mu_0) \leq D_\alpha((1-p)\mathcal{N}_\sigma((1-\alpha_{\mathbf{e}}^t)\Delta_{\mathbf{e}}^t|\mathcal{H}^t) + p\mathcal{N}_\sigma((1-\alpha_{\mathbf{e}}^t)\Delta_{\mathbf{e}}^t + \frac{2G'}{B})\|\mu_0|\mathcal{H}^t),
\tag{21}
$$

*where $p = \frac{B}{N}$ be the sub-sampling rate of the minibatch, $G' = \max\{0, G + \sigma - C_1\}$ and $\alpha_{\mathbf{e}}^t$ defined in (9).*

The proof is given in Appendix A.3.1. By applying the recursion of $\mathbf{e}^t$ given in Lemma A.6 to $D_\alpha(\mathcal{N}_\sigma(\Delta_{\mathbf{e}}^t)\|\mu_0|\mathcal{H}^t)$, we have:

$$
\begin{aligned}
D_\alpha(\mathcal{N}_\sigma(\Delta_{\mathbf{e}}^{t+1})\|\mu_0|\mathcal{H}^{t+1}) &\overset{(a)}{\leq} D_\alpha((1-p)\mathcal{N}_\sigma((1-\alpha_{\mathbf{e}}^t)\Delta_{\mathbf{e}}^t) + p\mathcal{N}_\sigma((1-\alpha_{\mathbf{e}}^t)\Delta_{\mathbf{e}}^t + \frac{2G'}{B})\|\mu_0|\mathcal{H}^{t+1}) \\
&\overset{(b)}{\leq} (1+\alpha_{\mathbf{e}}^t)D_\alpha(\mathcal{N}_\sigma((1-\alpha_{\mathbf{e}}^t)\Delta_{\mathbf{e}}^t)\|\mu_0|\mathcal{H}^{t+1}) \\
&\quad + (1+\frac{1}{\alpha_{\mathbf{e}}^t})D_\alpha((1-p)\mathcal{N}_\sigma((1-\alpha_{\mathbf{e}}^t)\Delta_{\mathbf{e}}^t) + p\mathcal{N}_\sigma((1-\alpha_{\mathbf{e}}^t)\Delta_{\mathbf{e}}^t + \frac{2G'}{B})\|\mathcal{N}_\sigma((1-\alpha_{\mathbf{e}}^t)\Delta_{\mathbf{e}}^t)|\mathcal{H}^{t+1}) \\
&\overset{(c)}{=} (1+\alpha_{\mathbf{e}}^t)D_\alpha(\mathcal{N}_\sigma((1-\alpha_{\mathbf{e}}^t)\Delta_{\mathbf{e}}^t)\|\mu_0|\mathcal{H}^{t+1}) \\
&\quad + (1+\frac{1}{\alpha_{\mathbf{e}}^t})D_\alpha((1-p)\mu_0 + p\mathcal{N}_\sigma(\frac{2G'}{B})\|\mu_0) \\
&\overset{(d)}{\leq} (1-\alpha_{\mathbf{e}}^t)D_\alpha(\mathcal{N}_\sigma(\Delta_{\mathbf{e}}^t)\|\mu_0|\mathcal{H}^{t+1}) + (1+\frac{1}{\alpha_{\mathbf{e}}^t})D_\alpha(p\mathcal{N}_\sigma(\frac{2G'}{B}) + (1-p)\mu_0\|\mu_0) \\
&\overset{(e)}{\leq} (1-\alpha_{\mathbf{e}}^t)D_\alpha(\mathcal{N}_\sigma(\Delta_{\mathbf{e}}^t)\|\mu_0|\mathcal{H}^{t+1}) + (1+\frac{1}{\alpha_{\mathbf{e}}^t})\frac{8p^2\alpha G'^2}{\sigma_1^2 B^2}
\end{aligned}
\tag{22}
$$

where $(a)$ applies Lemma A.6; $(b)$ applies Lemma A.8 and choose $\beta = \alpha_{\mathbf{e}}^t$; $(c)$ shifts the mean of the Gaussian distributions by $(1-\alpha_{\mathbf{e}}^t)\Delta_{\mathbf{e}}^t$ in the second term; $(d)$ applies Corollary A.9 to move the factor $(1-\alpha_{\mathbf{e}}^t)$ in the first term outside the Rényi divergence, and notice

$$
(1-\alpha_{\mathbf{e}}^t)^2(1+\alpha_{\mathbf{e}}^t) = (1-(\alpha_{\mathbf{e}}^t)^2)(1-\alpha_{\mathbf{e}}^t) \leq (1-\alpha_{\mathbf{e}}^t);
$$

$(e)$ applies Lemma A.10 to the second term. Towards this end, we have already derived the change of Rényi divergence for the one-step update of $\Delta_{\mathbf{e}}^t$. Then, we further recursively expand $\Delta_{\mathbf{e}}^t$ to $\Delta_{\mathbf{e}}^0$ and notice $\Delta_{\mathbf{e}}^0 = 0$ and we have:

$$
8(\alpha_{\mathbf{e}}^{t+1})^2 D_\alpha(\mathcal{N}_\sigma(\Delta_{\mathbf{e}}^{t+1})\|\mu_0) \overset{(a)}{\leq} (\alpha_{\mathbf{e}}^{t+1})^2 \frac{8p^2\alpha G'^2}{\sigma_1^2 B^2} \sum_{\tau=0}^t (1+\frac{1}{\alpha_{\mathbf{e}}^\tau}) \prod_{\tau_1=\tau+1}^t (1-\alpha_{\mathbf{e}}^{\tau_1})
$$

$$\overset{(b)}{\leq} \frac{8p^2\alpha G'^2}{\sigma_1^2 B^2} \sum_{\tau=0}^{t} (\alpha_{\mathbf{e}}^{t+1})^2 (\frac{1+\alpha_{\mathbf{e}}^{\tau}}{\alpha_{\mathbf{e}}^{\tau}}) \prod_{\tau_1=\tau+1}^{t} \frac{\tau_1 \max\{0, G'-C_2\}}{C_2 + \tau_1 \max\{0, G'-C_2\}}$$

$$\overset{(c)}{\leq} \frac{8p^2\alpha}{\sigma_1^2} \sum_{\tau=0}^{t} (\frac{G'^2 C_2 (2C_2 + \tau \max\{0, G'-C_2\})}{B^2(C_2 + \max\{0, (t-1)(G'-C_2)\})^2}) \prod_{\tau_1=\tau+1}^{t} \frac{\tau_1 \max\{0, G'-C_2\}}{C_2 + \tau_1 \max\{0, G'-C_2\}}$$

$$\overset{(d)}{\leq} \frac{8p^2\alpha}{\sigma_1^2} \sum_{\tau=0}^{t} (\frac{G'^2 C_2 (2C_2 + \tau \max\{0, G'-C_2\})}{B^2(C_2 + \max\{0, (t-1)(G'-C_2)\})^2}) \left(\frac{t \max\{0, G'-C_2\}}{C_2 + t \max\{0, G'-C_2\}}\right)^{t-\tau}$$

$$\overset{(e)}{\leq} \frac{8p^2\alpha \min\{C_2^2, G'^2/B^2\}}{\sigma_1^2} \frac{2C_2 + (t+1)\max\{0, G'-C_2\}}{C_2 + \max\{0, (t-1)(G'-C_2)\}}$$

$$= \frac{8p^2\alpha \min\{C_2^2 B^2, G'^2\}}{\sigma_1^2 B^2} \frac{2 + (t+1)\frac{\max\{G'-C_2, 0\}}{C_2}}{1 + \frac{\max\{(t-1)(G'-C_2), 0\}}{C_2}}(t-1) \tag{23}$$

where $(a)$ expand $\Delta_{\mathbf{e}}^t$ to $\Delta_{\mathbf{e}}^0$; $(b), (c)$ substitute the bound of $\alpha_{\mathbf{e}}^t$ in (9); $(d)$ relaxes $\frac{\tau_1 \max\{0, G'-C_2\}}{C_2 + \tau_1 \max\{0, G'-C_2\}} \leq \frac{t \max\{0, G'-C_2\}}{C_2 + t \max\{0, G'-C_2\}}, \forall \tau_1 \leq t$; $(e)$ sums over $\tau$ and bound

$$\sum_{\tau=0}^{t} \tau \left(\frac{t \max\{0, G'-C_2\}}{C_2 + t \max\{0, G'-C_2\}}\right)^{t-\tau} \leq \frac{1+t}{1 - \left(\frac{t \max\{0, G'-C_2\}}{C_2 + t \max\{0, G'-C_2\}}\right)}$$

and

$$\sum_{\tau=0}^{t} \left(\frac{t \max\{0, G'-C_2\}}{C_2 + t \max\{0, G'-C_2\}}\right)^{t-\tau} \leq \frac{1}{1 - \left(\frac{t \max\{0, G'-C_2\}}{C_2 + t \max\{0, G'-C_2\}}\right)}.$$

**Bounding the second term in** (20): Next, let us bound the second term in (20) by directly applying Lemma A.10. More specifically,

$$D_\alpha(\mathcal{N}_\sigma(\Delta_{\mathbf{g}}^t)\|\mu_0|\mathcal{H}^t) = D_\alpha(\mathcal{N}_\sigma(\frac{1}{B}\sum_{i\in\mathcal{B}^t} \text{clip}\,(\mathbf{g}_i^t, C_1))\|\mathcal{N}_\sigma(\frac{1}{B}\sum_{i\in\mathcal{B}'} \text{clip}\,(\mathbf{g}_i^t, C_1))|\mathcal{H}^t)$$

which is the difference between the sub-sampled Gaussian mechanism denoted as $\mathcal{M}(\mathcal{D}) = \frac{1}{B}\sum_{i\in\mathcal{B}^t} \text{clip}\,(\mathbf{g}_i^t, C_1) + \mathbf{w}^t$ and $\mathcal{M}(\mathcal{D}') = \frac{1}{B}\sum_{i\in\mathcal{B}'} \text{clip}\,(\mathbf{g}_i^t, C_1) + \mathbf{w}^t$. The sensitivity of the subsampled Gaussian mechanism is $\frac{2C_1}{B}$. By choosing $\sigma > \frac{8C_1}{B}$, and $p \leq \frac{1}{5}$, we directly apply Lemma A.10 and obtain

$$D_\alpha(\mathcal{M}(\mathcal{D})\|\mathcal{M}(\mathcal{D}')) = D_\alpha(\mathcal{N}_\sigma(\Delta_{\mathbf{g}}^t)\|\mu_0)$$
$$\overset{(a)}{\leq} D_\alpha((1-p)\mu_0 + p\mathcal{N}_\sigma(\frac{2C_1}{B})\|\mu_0) \overset{(b)}{\leq} \frac{8p^2 C_1^2 \alpha}{B^2 \sigma_1^2}, \tag{24}$$

where $(a)$ and $(b)$ directly applies (25) in Lemma A.10.

By substituting the bound of the two terms (23), (24) to (20), we obtain:

$$D_\alpha(\mathcal{A}_1(\mathbf{x}^t, \mathcal{D})\|\mathcal{A}_1(\mathbf{x}^t, \mathcal{D}')|\mathcal{H}^t) = D_\alpha(\mathcal{N}_\sigma(\mathbf{v}^t - \mathbf{v}'^t)\|\mu_0|\mathcal{H}^t)$$
$$\leq 2D_\alpha(\mathcal{N}_\sigma(\Delta_{\mathbf{e}}^t)\|\mu_0|\mathcal{H}^t) + 2D_\alpha(\mathcal{N}_\sigma(\Delta_{\mathbf{g}}^t)\|\mu_0|\mathcal{H}^t)$$
$$\leq \frac{(\frac{\max\{G'-C_2, 0\}}{C_2}(t+1) + 2)}{(\frac{\max\{(G'-C_2)(t-1), 0\}}{C_2} + 1)} \frac{16p^2\alpha \min\{C_2^2 B^2, G'^2\}}{\sigma_1^2 B^2} + \frac{16p^2 C_1^2 \alpha}{B^2 \sigma_1^2}.$$

Therefore, by the definition of RDP (Definition A.4), $\mathcal{A}_1^t$ guarantees $(\alpha, \epsilon^t)$-RDP where

$$\epsilon^t = \frac{16\alpha}{\sigma_1^2 N^2} \cdot \left(C_1^2 + \frac{(\frac{\max\{G'-C_2, 0\}}{C_2}(t+1) + 2)}{(\frac{\max\{(G'-C_2)(t-1), 0\}}{C_2} + 1)} \min\{C_2^2, G'^2\}\right).$$

Substitute the above bound to Step II, we have $\mathcal{H}^t$ satisfies $(\alpha, \sum_{\tau=0}^{t} \epsilon^\tau)$-RDP, where

$$\sum_{\tau=0}^{t} \epsilon^\tau = \sum_{\tau=0}^{t} \frac{16\alpha}{\sigma_1^2 N^2} \cdot \left(C_1^2 + \frac{(\frac{\max\{G'-C_2, 0\}}{C_2}(\tau+1) + 2)}{(\frac{\max\{(G'-C_2)(\tau-1), 0\}}{C_2} + 1)} \min\{C_2^2 B^2, G'^2\}\right)$$

$$= \frac{16\alpha t C_1^2}{\sigma_1^2 N^2} + \frac{16\alpha \min\{C_2^2 B^2, G'^2\}}{\sigma_1^2 N^2} \sum_{\tau=0}^{t} \left(1 + \frac{\frac{2\max\{G'-C_2,0\}}{C_2} + 1}{\left(\frac{\max\{(G'-C_2)(\tau-1),0\}}{C_2} + 1\right)}\right)$$

$$\overset{(a)}{\leq} \frac{16\alpha t C_1^2}{\sigma_1^2 N^2} + \frac{32t\alpha \min\{C^2, G'^2\}}{\sigma_1^2 N^2},$$

where $(a)$ bounds $\sum_{\tau=0}^{t} \left(1 + \frac{\frac{2\max\{G'-C_2,0\}}{C_2}+1}{\left(\frac{\max\{(G'-C_2)(\tau-1),0\}}{C_2}+1\right)}\right)$ by $2t$ and bounds $C_2$ by $C_2 \leq C/B$.

Therefore, by choosing $\sigma_1^2 \geq \frac{32T(C_1^2 + 2\min\{C^2, G'^2\})\log(1/\delta)}{N^2 \epsilon^2}$, DiceSGD guarantees $(\epsilon, \delta)$-DP for $T$ iterations. The proof of the theorem is completed. ∎

### A.3 ADDITIONAL LEMMAS

**Lemma A.7** (Proposition B.4.10. Gil (2011)). *The Rényi divergence between two Gaussian distributions with the same variance $\mathcal{N}(a, \sigma^2), \mathcal{N}(b, \sigma^2)$ is*

$$D_\alpha(\mathcal{N}(a, \sigma^2)\|\mathcal{N}(b, \sigma^2)) = \frac{\alpha(a-b)^2}{2\sigma^2}.$$

*Proof.* By definition of Rényi divergence and Gaussian distribution, we have:

$$
\begin{aligned}
D_\alpha(\mathcal{N}(a, \sigma^2)\|\mathcal{N}(b, \sigma^2)) &= \frac{1}{\alpha-1}\log\left(\frac{1}{\sqrt{2\pi\sigma^2}}\int_x \left(\exp(-(x-a)^2/2\sigma^2)\right)^\alpha \left(\exp(-(x-b)^2/2\sigma^2)\right)^{1-\alpha} \mathrm{d}x\right) \\
&= \frac{1}{\alpha-1}\log\left(\frac{1}{\sqrt{2\pi\sigma^2}}\int_x \exp\left(-\frac{\alpha(x-a)^2+(1-\alpha)(x-b)^2}{2\sigma^2}\right)\mathrm{d}x\right) \\
&= \frac{1}{\alpha-1}\log\left(\frac{1}{\sqrt{2\pi\sigma^2}}\int_x \exp\left(-\frac{(x-(\alpha a+(1-\alpha)b))^2+\alpha(1-\alpha)(a-b)^2}{2\sigma^2}\right)\mathrm{d}x\right) \\
&= \frac{1}{\alpha-1}\log\left(\exp\left(-\frac{\alpha(1-\alpha)(a-b)^2}{2\sigma^2}\right)\frac{1}{\sqrt{2\pi\sigma^2}}\int_x \exp\left(-\frac{(x-(\alpha a+(1-\alpha)b))^2}{2\sigma^2}\right)\mathrm{d}x\right) \\
&= \frac{1}{\alpha-1}\left(-\frac{\alpha(1-\alpha)(a-b)^2}{2\sigma^2}+\log\left(\frac{1}{\sqrt{2\pi\sigma^2}}\int_x \exp\left(-\frac{(x-(\alpha a+(1-\alpha)b))^2}{2\sigma^2}\right)\mathrm{d}x\right)\right) \\
&\overset{(a)}{=} \frac{1}{\alpha-1}\left(-\frac{\alpha(1-\alpha)(a-b)^2}{2\sigma^2}+\log(1)\right) \\
&= \frac{\alpha(a-b)^2}{2\sigma^2},
\end{aligned}
$$

where in $(a)$, we notice the second term is the PDF of $\mathcal{N}(\alpha a + (1-\alpha)b, \sigma^2)$, so its integral is 1. This completes the proof for the lemma. $\square$

**Lemma A.8.** *Given three Gaussian distributions $\mathcal{N}(a, \sigma^2), \mathcal{N}(b, \sigma^2), \mathcal{N}(c, \sigma^2)$, and constant $\beta > 0$, we have that*

$$D_\alpha(\mathcal{N}(a, \sigma^2)\|\mathcal{N}(b, \sigma^2)) \leq (1+\beta)D_\alpha(\mathcal{N}(a, \sigma^2)\|\mathcal{N}(c, \sigma^2)) + (1+\frac{1}{\beta})D_\alpha(\mathcal{N}(c, \sigma^2)\|\mathcal{N}(b, \sigma^2)).$$

*Proof.* Directly apply Lemma A.7, we have

$$
\begin{aligned}
D_\alpha(\mathcal{N}(a, \sigma^2)\|\mathcal{N}(b, \sigma^2)) &= \frac{\alpha(a-b)^2}{2\sigma^2} \\
&\overset{(5)}{\leq} \frac{\alpha\left((1+\beta)(a-c)^2+(1+\frac{1}{\beta})(c-b)^2\right)}{2\sigma^2} \\
&= (1+\beta)D_\alpha(\mathcal{N}(a, \sigma^2)\|\mathcal{N}(c, \sigma^2)) + (1+\frac{1}{\beta})D_\alpha(\mathcal{N}(c, \sigma^2)\|\mathcal{N}(b, \sigma^2)).
\end{aligned}
$$

The proof is completed. $\square$

**Corollary A.9.** *For the Rényi divergence between $\mathcal{N}(a, \sigma^2)$ and $\mathcal{N}(0, \sigma^2)$, we have*

$$D_\alpha(\mathcal{N}(a, \sigma^2)\|\mathcal{N}(0, \sigma^2)) = a^2 D_\alpha(\mathcal{N}(1, \sigma^2)\|\mathcal{N}(0, \sigma^2)).$$

*Proof.* Directly applies Lemma A.7 for the special case $b = 0$, the corollary is proved. $\square$

**Lemma A.10** (Theorem 11 Mironov et al. (2019)). *If $p \leq \frac{1}{5}, \sigma > 4C$ and $\alpha$ satisfies*

$$1 \leq \alpha \leq \frac{1}{2}\sigma^2 C_3 - 2\ln\sigma, \text{ and } \alpha \leq \frac{\frac{1}{2}\sigma^2 C_3^2 - \ln 5 - 2\ln\sigma}{C_3 + \ln(p\alpha) + 1/(2\sigma^2)},$$

where $C_3 = 1 + \frac{1}{p(\alpha-1)}$, then the sub-sampled Gaussian mechanism $\mathcal{M}$ applied to a function of $\ell_2$-sensitivity $C$ with the sub-sampling rate $p$ satisfies

$$D_\alpha(\mathcal{M}(\mathcal{D})\|\mathcal{M}(\mathcal{D}')) \leq D_\alpha((1-p)\mathcal{N}_\sigma(0, C^2\sigma^2\cdot I)+p\mathcal{N}_\sigma(C, C^2\sigma^2\cdot I)\|\mathcal{N}_\sigma(0, C^2\sigma^2\cdot I)) \leq \frac{2p^2C^2\alpha}{\sigma^2}. \tag{25}$$

Therefore, it satisfies $(\alpha, \epsilon)$-RDP where $\epsilon = \frac{2p^2C^2\alpha}{\sigma^2}$.

### A.3.1 PROOF FOR LEMMA A.6

*Proof.* First note that using the update rule of $\mathbf{e}^t$ in Algorithm 2, we have

$$\mathbf{e}^{t+1} = \mathbf{e}^t + \frac{1}{B}\sum_{i\in\mathcal{B}^t}\nabla f(\mathbf{x}^t;\xi_i) - \mathbf{v}^t, \quad \mathbf{e}'^{t+1} = \mathbf{e}'^t + \frac{1}{B}\sum_{i\in\mathcal{B}'^t}\nabla f(\mathbf{x}^t;\xi_i) - \mathbf{v}'^t.$$

Recall $\mathcal{D}$ and $\mathcal{D}'$ differs by a single sample, and let $\xi'$ denote this particular sample in $\mathcal{D}'$. That is, $\mathcal{D}' = \mathcal{D} \cup \{\xi'\}$. Recall $\mathcal{B}$ is a random subset of $\mathcal{D}$ where each element is independently selected with probability $p$. Similarly, $\mathcal{B}'$ is a random subset of $\mathcal{D}'$, and with probability $p$, $\mathcal{B}'$ samples $\xi'$; and with probability $1-p$, $\mathcal{B}'$ does not sample $\xi'$. Then taking expectation with respect to $\mathcal{B}, \mathcal{B}'$, the mean of $\mathbf{e}^{t+1}, \mathbf{e}'^{t+1}$ follows

$$\mathbb{E}_{\mathcal{B}^t}[\mathbf{e}^{t+1}] = \mathbb{E}_{\mathcal{B}^t}\left[\mathbf{e}^t + \frac{1}{B}\sum_{i\in\mathcal{B}^t}\nabla f(\mathbf{x}^t;\xi_i) - \mathbf{v}^t\right]$$

$$\mathbb{E}_{\mathcal{B}'^t}[\mathbf{e}'^{t+1}] = \mathbb{E}_{\mathcal{B}'^t}\left[\mathbf{e}'^t + \frac{1}{B}\sum_{i\in\mathcal{B}'^t}\nabla f(\mathbf{x}^t;\xi_i) - \mathbf{v}'^t\right]$$

$$= \mathbf{e}'^t + \mathbb{E}_{\mathcal{B}'^t}\left[\mathbb{E}_{\xi'}\left[\frac{1}{B}\sum_{i\in\mathcal{B}'^t}\nabla f(\mathbf{x}^t;\xi_i) - \mathbf{v}'^t\right]\right] \tag{26}$$

$$= \mathbb{E}_{\mathcal{B}^t}\left[(1-p)(\mathbf{e}'^t + \frac{1}{B}\sum_{i\in\mathcal{B}^t}\nabla f(\mathbf{x}^t;\xi_i) - \mathbf{v}'^t) + p(\mathbf{e}'^t + \frac{1}{B}\sum_{i\in\mathcal{B}^t\cup\{\xi'\}}\nabla f(\mathbf{x}^t;\xi_i) - \mathbf{v}'^t)\right]$$

Recall that $\Delta_{\mathbf{e}}^{t+1} = \mathbf{e}^{t+1} - \mathbf{e}'^{t+1}$. Then by the quasi-convexity of Rényi divergence, we have

$$D_\alpha(\mathcal{N}_\sigma(\Delta_{\mathbf{e}}^{t+1})\|\mu_0) = D_\alpha(\mathcal{N}_\sigma(\mathbf{e}'^{t+1})\|\mathcal{N}_\sigma(\mathbf{e}^{t+1}))$$

$$\overset{(a)}{\leq} 2D_\alpha\bigg(\mathbb{E}_{\mathcal{B}^t}\left[p\mathcal{N}_\sigma(\mathbf{e}'^t + \frac{1}{B}\sum_{i\in\mathcal{B}^t\cup\{\xi'\}}\nabla f(\mathbf{x}^t;\xi_i) - \mathbf{v}'^t) + (1-p)\mathcal{N}_\sigma(\mathbf{e}'^t + \frac{1}{B}\sum_{i\in\mathcal{B}^t}\nabla f(\mathbf{x}^t;\xi_i) - \mathbf{v}'^t)\right]$$

$$\left\|\mathbb{E}_{\mathcal{B}^t}\left[\mathcal{N}_\sigma(\mathbf{e}^t + \frac{1}{B}\sum_{i\in\mathcal{B}^t}\nabla f(\mathbf{x}^t;\xi_i) - \mathbf{v}^t, \sigma^2\cdot I)\right]\right\|\bigg)$$

$$\overset{(b)}{\leq} \sup_{\mathcal{B}^t} D_\alpha\bigg(p\mathcal{N}_\sigma(\mathbf{e}'^t + \frac{1}{B}\sum_{i\in\mathcal{B}^t\cup\{\xi'\}}\nabla f(\mathbf{x}^t;\xi_i) - \mathbf{v}'^t) + (1-p)\mathcal{N}_\sigma(\mathbf{e}'^t + \frac{1}{B}\sum_{i\in\mathcal{B}^t}\nabla f(\mathbf{x}^t;\xi_i) - \mathbf{v}^t)$$

$$\left\|\mathcal{N}_\sigma(\mathbf{e}^t + \frac{1}{B}\sum_{i\in\mathcal{B}^t}\nabla f(\mathbf{x}^t;\xi_i) - \mathbf{v}^t, \sigma^2\cdot I)\right\|\bigg)$$

$$\overset{(c)}{=} \sup_{\mathcal{B}^t} D_\alpha\bigg(p\mathcal{N}_\sigma((1-\alpha_{\mathbf{e}}^t)\Delta_{\mathbf{e}}^t + \frac{1}{B}\sum_{i\in\mathcal{B}^t\cup\{\xi'\}}(\nabla f(\mathbf{x}^t;\xi_i) - \text{clip}(\nabla f(\mathbf{x}^t;\xi_i), C_1))$$

$$- \frac{1}{B}\sum_{i\in\mathcal{B}^t}(\nabla f(\mathbf{x}^t;\xi_i) - \text{clip}(\nabla f(\mathbf{x}^t;\xi_i), C_1)))$$

$$+ (1-p)\mathcal{N}_\sigma((1-\alpha_{\mathbf{e}}^t)\Delta_{\mathbf{e}}^t)\bigg\|\mu_0\bigg)$$

$$\overset{(d)}{\leq} D_\alpha\bigg(p\mathcal{N}_\sigma((1-\alpha_{\mathbf{e}}^t)\Delta_{\mathbf{e}}^t + \frac{1}{B}\left(\nabla f(\mathbf{x}^t;\xi') - \text{clip}(\nabla f(\mathbf{x}^t;\xi'), C_1)\right))$$

$$+ (1-p)\mathcal{N}_\sigma((1-\alpha_{\mathbf{e}}^t)\Delta_{\mathbf{e}}^t)\Big\|\mu_0\Big)$$

$$\overset{(e)}{\leq} D_\alpha\Big((1-p)\mathcal{N}_\sigma((1-\alpha_{\mathbf{e}}^t)\Delta_{\mathbf{e}}^t) + p\mathcal{N}_\sigma((1-\alpha_{\mathbf{e}}^t)\Delta_{\mathbf{e}}^t + \frac{2G'}{B})\Big\|\mu_0\Big)$$

where $(a), (b)$ uses the quasi-convexity of Rényi divergence that

$$D_\alpha(\mathbb{E}_\theta[P(\theta)]\| \mathbb{E}_\theta[Q(\theta)]) \leq \max_\theta\{D_\alpha(P(\theta)\|Q(\theta))\}$$

with $\theta = \mathcal{B}$; $(c)$ shifts the mean of Gaussian distributions by $(1-\alpha_{\mathbf{e}}^t)\mathbf{e}^t + \frac{1}{B}\sum_{i\in\mathcal{B}^t}\nabla f(\mathbf{x}^t;\xi_i) - \mathbf{v}^t$; $(d)$ cancels the identical terms in $\mathcal{B}$; $(e)$ bounds $\frac{1}{B}(\nabla f(\mathbf{x}^t;\xi') - \mathrm{clip}(\nabla f(\mathbf{x}^t;\xi'),C_1))$ by its sensitivity $\frac{2G'}{B}$.

$\square$

## A.4   PROOF OF LEMMA A.1

For the first part of the lemma, by definition, we have:

$$\mathrm{Var}(c(X)) = \mathbb{E}\|c(X) - \mathbb{E}[c(X)]\|^2$$
$$\overset{(a)}{\leq} \mathbb{E}\|c(X) - c(\mathbb{E}[X])\|^2 \tag{27}$$
$$\overset{(b)}{\leq} \mathbb{E}\|X - \mathbb{E}[X]\|^2 = \mathrm{Var}(X),$$

where $(a)$ uses the fact that $\mathbb{E}(X - \mathbb{E}[X])^2 \leq \mathbb{E}(X - Y)^2, \forall Y$; $(b)$ applies the fact that $c(\cdot)$ is a non-expansive mapping, so that $\|c(a) - c(b)\|^2 \leq \|a - b\|^2$.

For the second part of the lemma, we notice that clipping operation is a projection operation to a convex set, so it is non-expansive (Takahashi, 1970),

$$\mathrm{clip}(x, C) = \arg\min_{\|z\|\leq C} \frac{1}{2}\|x - z\|^2,$$

where set $\{x|\|x\| \leq C\}$ is convex. Therefore, $c(x) = x - \mathrm{clip}(x, C)$ is also non-expansive (Takahashi, 1970) that

$$\|c(x) - c(y)\| \leq \|x - y\|, \forall x, y.$$

# B   PROOF OF RESULTS IN SECTION 2

## B.1   EXAMPLE OF CONSTANT CLIPPING BIAS

For any fixed positive clipping threshold $C$, let us consider the following function

$$f(x, \xi) = \begin{cases} -C'(x - \xi + \frac{C'}{2}), & x \leq \xi - C' \\ \frac{1}{2}(x - \xi)^2, & |x - \xi| \leq C' \\ C'(x - \xi - \frac{C'}{2}), & x \geq \xi + C' \end{cases},$$

with $C' = \lceil C \rceil + 1$. The per-sample gradient of this problem is

$$\nabla f(x, \xi) = \begin{cases} -C', & x \leq \xi - C' \\ x - \xi, & |x - \xi| \leq C' \\ C', & x \geq \xi + C' \end{cases}.$$

By setting the dataset size as $N = C' + 1$, and the samples are $\xi_i = -1, \forall i = 1, \ldots, C'$, and $\xi_{C'+1} = C'$, we can verify that $f(x)$ satisfies Assumptions 3.1-3.5 with certain constants.

Next, we analyze the stationary solution to this problem with and without clipping operation, i.e., the expected stationary solution of SGD and Clipped SGD. The stationary solution of SGD is $\nabla f(x^\star) = 0$, so $\sum_{i=1}^N (x^\star - \xi_i) = 0$, $x^\star = 0$. On the other hand, by running clipped SGD with clipping threshold $C$, the stationary solution is $\tilde{x}^\star = \frac{C'}{C'+C} \geq \frac{1}{2}$. This indicates that with a small enough clipping threshold $C$, clipped SGD converges to a neighborhood of the stationary solution of the problem, with an $\mathcal{O}(1)$ clipping bias.

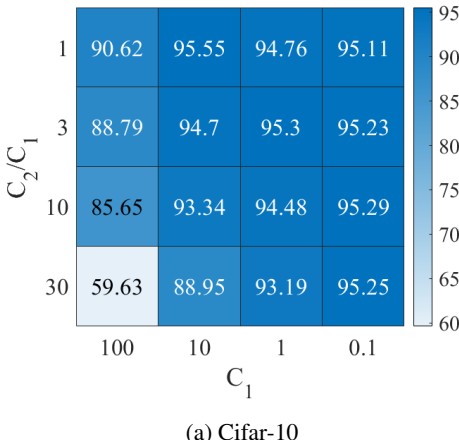 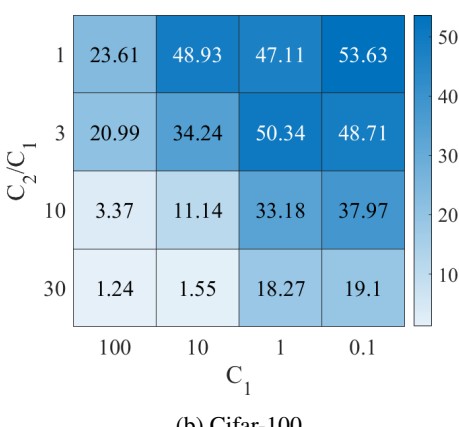

| (a) Cifar-10 | (b) Cifar-100 |

Figure 2: The testing accuracy for Cifar-10 and Cifar-100 trained with DiceSGD under different $C_1, C_2$ with fixed effective stepsize.

## C   ADDITIONAL EXPERIMENTS

### C.1   ABLATION STUDY

In this section, we provide the ablation study on the choices of the clipping threshold $C_1, C_2$ and the learning rate $\eta$. The experiments are conducted on the Cifar-10 and Cifar-100 datasets, with fixed $(2, 10^{-5})$-DP. We fine-tune the ViT-small model for 2 epochs with batch size $B = 1000$, i.e., $T = \frac{2 \times 50000}{1000} = 100$.

In the experiment, we first use a re-parameterization method in De et al. (2022) to fix the product of the step size $\eta$ and the clipping threshold $C_1$. By doing this, we normalize the update $\mathbf{v}^t + \mathbf{w}^t$ ny the clipping threshold $C_1$ and fix the "effective stepsize" of the algorithm, i.e.,

$$\mathbf{x}^{t+1} = \mathbf{x}^t - \eta^t C_1 \left( \frac{\mathbf{v}^t + \mathbf{w}^t}{C_1} \right),$$

where $\mathbf{v}^t$ and $\mathbf{w}^t$ scales with $C_1$. We study the impact of different combinations of $C_1$ and $C_2$. We choose $C_1 = \{100, 10, 1, 0.1\}$ and $C_2 = \{1, 3, 10, 30\} \times C_1$, and the results are shown in Figure 2. From the figure, we see that choosing $C_2 = C_1$ gives the best performance in most cases. Additionally, choosing larger $C_1$ gives a worse result, and DiceSGD benefits from using a small clipping threshold.

Next, we fix $C_2 = C_1$ and study the impact of different combinations between the clipping threshold $C_1$ and stepsize $\eta$. We choose $C_1 = \{100, 10, 1, 0.1\}$ and $\eta = \{3.0, 1.0, 0.3, 0.1, 0.03, 0.01\}/C_1$. The result is shown in Figure 3. From the figure, we see that DiceSGD benefits from using a small clipping threshold, and exists a best $\eta \times C_1$ that gives the best performance.

### C.2   TRAINING GPT-2

We train a GPT-2 model with the E2E dataset, which contains template-like information in the restaurant domain to be mapped to natural language with end-to-end training that has 42000 training samples. The GPT model is fine-tuned for 10 epochs, with batch size $B = 1000$, so $T = \frac{42000 \times 10}{1000} = 420$. We use the similar AdamW variant of DPSGD and DiceSGD for training and set initial stepsize $\eta^0 = 2 \times 10^{-3}$ with learning rate warm-up and linear decay. The algorithm is guaranteed $(8, 8 \times 10^{-6})$-DP. We report the testing loss for each epoch in Figure 4.

**Ablation study for GPT-2**   In this section, we provide the ablation study on the choices of the clipping threshold $C_1$ and the learning rate $\eta$ with GPT-2. We fine-tune the GPT-2 model for 5 epochs with batch size $B = 1000$, i.e., $T = \frac{5 \times 42000}{1000} = 210$. We use the AdamW variant of DiceSGD for training with learning rate warm-up and linear decay. The algorithm is guaranteed $(8, 8 \times 10^{-6})$-DP. We fix $C_2 = C_1$ and study the impact of different combinations between the clipping threshold $C_1$

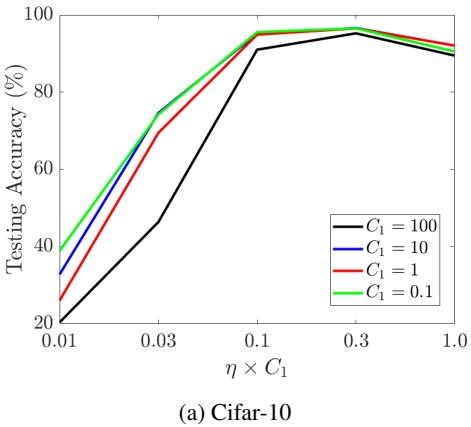
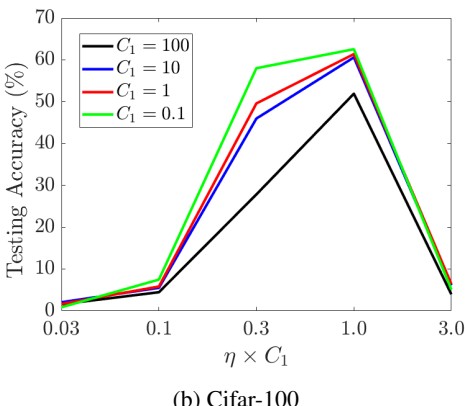

(a) Cifar-10 (b) Cifar-100

Figure 3: The testing accuracy for Cifar-10 and Cifar-100 trained with DiceSGD under different $C_1, \eta$ with fixed $C_2 = C_1$.

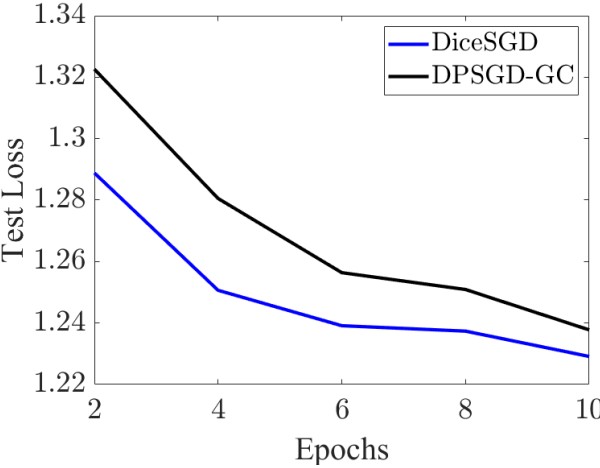

Figure 4: Testing loss of DPSGD and DiceSGD fine-tuning GPT-2 on E2E dataset, with clipping thresholds $C = C_1 = C_2 = 1$ and guarantees $(8, 8 \times 10^{-6})$-DP.

and stepsize $\eta$. We choose $C_1 = \{100, 10, 1, 0.1\}$ and $\eta = \{2, 1, 0.5\} \times \{10^{-2}, 10^{-3}\}$. We report the testing loss for each combination of the initial stepsize $\eta$ and clipping threshold $C_1$ in Figure 5. From the result, we see that using a smaller clipping threshold gives a better result for DiceSGD.

### C.3 ADAM VARIANT OF DICESGD

In this section, we provide detailed updates of the Adam variant of DiceSGD in Algorithm 3.

---

**Algorithm 3** Adam variant of DiceSGD Algorithm

1: **Input:** $\mathbf{x}^0, \mathcal{D}, C_1, C_2, \eta, \beta_1, \beta_2, \epsilon_1$
2: **Initialize:** $\mathbf{e}^0 = 0, \mathbf{m}_1^0 = 0, \mathbf{m}_2^0 = 0$
3: **for** $t = 0, \ldots, T - 1$ **do**
4:    Randomly draw minibatch $\mathcal{B}^t$ from $\mathcal{D}$
5:    $\mathbf{v}^t = \frac{1}{B} \sum_{i \in \mathcal{B}^t} \text{clip}\left(\nabla f(\mathbf{x}^t; \xi_i), C_1\right) + \text{clip}\left(\mathbf{e}^t, C_2\right) + \mathbf{w}^t$, where $\mathbf{w}^t \sim \mathcal{N}(0, \sigma_1^2 \cdot \mathbf{I})$
6:    $\mathbf{m}_1^t = \beta_1 \mathbf{m}_1^{t-1} + (1 - \beta_1)\mathbf{v}^t$      // Update first-order moment estimate
7:    $\mathbf{m}_2^t = \beta_2 \mathbf{m}_2^{t-1} + (1 - \beta^2)(\mathbf{v}^t)^2$      // Update second-order moment estimate
8:    $\mathbf{x}^{t+1} = \mathbf{x}^t - \eta^t \frac{\mathbf{m}^t/(1-(\beta_1)^t)}{\sqrt{\mathbf{m}_2^t/(1-(\beta_2)^t)+\epsilon_1}}$,
9:    $\mathbf{e}^{t+1} = \mathbf{e}^t + \frac{1}{B} \sum_{i \in \mathcal{B}^t} \nabla f(\mathbf{x}^t; \xi_i) - \left(\frac{1}{B} \sum_{i \in \mathcal{B}^t} \text{clip}\left(\nabla f(\mathbf{x}^t; \xi_i), C_1\right) + \text{clip}\left(\mathbf{e}^t, C_2\right)\right).$
10: **end for**

---

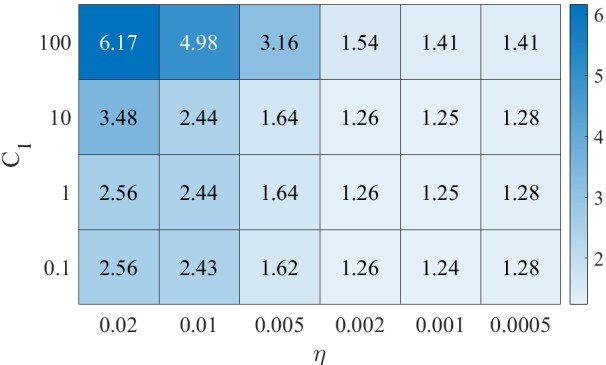

Figure 5: Testing loss (smaller the better) of DiceSGD on E2E dataset with different combinations of clipping thresholds and initial stepsizes.

## C.4 EFFICIENCY ANALYSIS

**Hyperparameter tuning efficiency** In our ablation study, we observe two patterns to achieve good accuracy: (1) $C_2$ needs to be close to $C_1$. (2) $C_1$ needs to be small. Therefore we can write $C_1 = C_2 = C$ and derive an automatic version of DiceSGD: in 2, $\mathbf{v}^t = \frac{1}{B} \sum_{i \in \mathcal{B}^t} \text{clip}\left(\nabla f(\mathbf{x}^t; \xi_i), C\right) + \text{clip}\left(\mathbf{e}^t, C\right)$ where $\text{clip}\left(x, C\right) = \frac{C}{\|x\|}$, known as the automatic clipping in Bu et al. (2024). Setting the injected noise $\sigma_1 = \sqrt{\frac{32T \cdot 3C^2 \log(1/\delta)}{N^2 \epsilon^2}} = \frac{\sqrt{96T \log(1/\delta)} C}{N \epsilon}$ satisfies DP. Let the learning rate $\eta^t$ absorbs $C$:

---

**Algorithm 4** Automatic DiceSGD Algorithm (without $C_1, C_2$)

1: **Input:** $\mathbf{x}^0, \mathcal{D}, \eta$
2: **Initialize:** $\mathbf{e}^0 = 0$
3: **for** $t = 0, \ldots, T-1$ **do**
4:     Randomly draw minibatch $\mathcal{B}^t$ from $\mathcal{D}$
5:     $\mathbf{v}^t = \frac{1}{B} \sum_{i \in \mathcal{B}^t} \text{clip}\left(\nabla f(\mathbf{x}^t; \xi_i), 1\right) + \text{clip}\left(\mathbf{e}^t, 1\right)$
6:     $\mathbf{x}^{t+1} = \mathbf{x}^t - \eta^t(\mathbf{v}^t + \mathbf{w}^t)$, where $\mathbf{w}^t \sim \frac{\sqrt{96T \log(1/\delta)}}{N \epsilon} \mathcal{N}(0, \mathbf{I})$
7:     $\mathbf{e}^{t+1} = \mathbf{e}^t + \frac{1}{B} \sum_{i \in \mathcal{B}^t} \nabla f(\mathbf{x}^t; \xi_i) - \mathbf{v}^t$.
8: **end for**

---

**Computational efficiency** In this part, let us briefly discuss the complexity of DiceSGD.

**Memory:** DiceSGD requires 3 times the memory of DPSGD-GC (which has similar time/space efficiency to the standard SGD), i.e., besides the summed clipped per-sample gradient, DiceSGD requires extra memory for both summed unclipped gradient and the feedback signal $\mathbf{e}^t$. **Computation:** The computation overhead of DiceSGD is minor compared with the cost of the per-sample clipped gradient computation. Specifically, the total computation of DiceSGD consists of $B \times$ per-sample (clipped) gradient computation $+2(B-1) \times$ gradient summation $+$ SGD update $+\mathbf{e}^t$ update, where DiceSGD requires extra $(B-1) \times$ gradient summation and one $\mathbf{e}^t$ update compared with DPSGD-GC. Additionally, in the distributed learning regime that is necessary to train large models, we expect that the gap in computational efficiency is insignificant, given that the communication cost acts as a dead weight (e.g. $\frac{\text{DP computation cost+communication cost}}{\text{non-DP computation cost+communication cost}} \approx 1.0$ if communication cost $\gg$ computation cost).

