3.4** (PL Condition). The gradient of the problem satisfies the following inequality:
$$\frac{1}{2\mu} \|\nabla f(\mathbf{x})\|^2 \geq f(\mathbf{x}) - f^\star, \forall \, \mathbf{x} \in \mathbb{R}^d.$$

The PL condition is also a widely used assumption when analyzing the convergence of SGD-type algorithms (Stich & Karimireddy, 2020; Lei et al., 2019; Karimi et al., 2016b). This assumption guarantees a quadratic growth to the gradient of the function. It is weaker than the strongly convex assumption as it only imposes a constraint based on the distance between one point to the set of optimal points.

**Assumption 3.5** (Bounded Variance). The stochastic gradient estimation is unbiased, i.e., $\mathbb{E}[\mathbf{g}] = \nabla f(\mathbf{x})$, and its variance satisfies that there exists a constant $\sigma$, such that $\mathbb{E} \|\nabla f(\mathbf{x}) - \mathbf{g}_i\|^2 \leq \frac{\sigma^2}{N}, \forall \, \mathbf{x} \in \mathbb{R}^d$.

**Assumption 3.6** (Bounded Gradient). The gradient of the function is bounded in the sense that there exists a positive constant $G = \sup_{\mathbf{x} \in \mathbb{R}^d} \|\nabla f(\mathbf{x})\| < \infty$.

Assumptions 3.5 and 3.6 are commonly used for analyzing clipping operation (Zhang et al., 2020; Qian et al., 2021; Song et al., 2020), the convergence of DP algorithms (Yang et al., 2022), and distributed optimization (Li et al., 2022; Zhang et al., 2022). Assumption 3.5 assumes a smaller variance compared with the typical assumption (i.e., $\mathbb{E} \|\nabla f(\mathbf{x}) - \mathbf{g}_i\|^2 \leq \sigma^2$), it implies that $\|\nabla f(\mathbf{x}) - \mathbf{g}_i\|^2 \leq \sigma^2, \forall \, i$, and it is necessary for bounding the clipping bias in the existing works (e.g.,in Yang et al. (2022)). Although these assumptions are also used in our analysis, contrasting with existing works, the clipping thresholds $C_1, C_2$ in DiceSGD do not depend on $G$ or $\sigma$.

We now present the convergence theorem of the proposed DiceSGD algorithm under the strong convexity condition 3.3.

**Theorem 3.7.** *Assume the problem satisfies Assumption 3.1, 3.2, 3.3, 3.5, and 3.6. Given any constant DP noise multiplier $\sigma_1$, by running DiceSGD (Algorithm 2) for $T$ iterations, choosing stepsize $\eta \leq \min\{\frac{1}{2\mu}, \frac{3}{32L}, \frac{\sqrt{C_2}}{5L\sqrt{C_2 + T \max\{0, G' - C_2\}}}\}$, $G' = \max\{0, G + \sigma - C_1\}$, clipping thresholds $C_2 \geq C_1 > 0$. It satisfies*

$$\mathbb{E}[\mathcal{L}^{T+1}] \leq (1 - \mu\eta)^T \mathcal{L}^0 + \frac{\eta L}{2\mu} \left( d\sigma_1^2 + \frac{5\sigma^2}{4B} + \frac{C_1^2}{2} \right) \tag{4}$$

*where we have defined*

$$\mathcal{L}^t := f(\tilde{\mathbf{x}}^t) - f^\star + \frac{\eta^3 L^2}{6} \|\mathbf{e}^t\|^2 + \frac{\eta(1 - L\eta)}{2} \|\nabla f(\mathbf{x}^t)\|^2,$$

*with $\tilde{\mathbf{x}}^t := \mathbf{x}^t - \eta \mathbf{e}^t$.*

**Proof sketch of Theorem 3.7:**

Table 1: The comparison between DPSGD, DPSGD-GC, and DiceSGDin terms of convergence, privacy noise, and clipping thresholds. ($\tilde{G} = 2C^2 + C_1^2$)

| Algorithm | Convergence Rate | Privacy Noise Variance | Assumptions | Clipping |
|---|---|---|---|---|
| DPSGD | $\mathcal{O}\left(\frac{G^2 \log(1/\delta)}{N^2 \epsilon^2}\right)$ | $\mathcal{O}(\frac{G^2 T \log(\frac{1}{\delta})}{N^2 \epsilon^2})$ | 3.3, 3.5, 3.6 | $C \geq G + \sigma$ |
| DPSGD-GC | $\mathcal{O}\left(\frac{C^2 \log(1/\delta)}{N^2 \epsilon^2}\right) + \mathcal{O}(1)$ | $\mathcal{O}(\frac{C^2 T \log(\frac{1}{\delta})}{N^2 \epsilon^2})$ | 3.3, 3.5, 3.6 | $C < G + \sigma$ |
| **DiceSGD** | $\mathcal{O}\left(\frac{\tilde{G} \log(N\epsilon) \log(1/\delta)}{N^2 \epsilon^2}\right)$ | $\mathcal{O}(\frac{\tilde{G} T \log(\frac{1}{\delta})}{N^2 \epsilon^2})$ | 3.3/3.4, 3.5, 3.6 | Arbitrary |

1. We first define a *virtual* sequence $\{\tilde{\mathbf{x}}^t\}$, which is updated with the "unclipped gradients", and we can apply the classic convergence analysis of SGD (without gradient clipping) for strongly convex problems on the virtual sequence. Due to the EF mechanism, the convergence result for DiceSGD has an extra term depending on $\mathbb{E}\left\|\mathbf{e}^t\right\|^2$.

2. With the update of $\mathbf{e}^t$, we can derive a recursive bound on the extra term $\mathbb{E}\left\|\mathbf{e}^t\right\|^2$ depending on $\left\|\mathbf{e}^{t-1}\right\|^2$ and the expected gradients. Unlike EF for contracting error which depends on the gradients with a constant factor independent of $T$, the error $\mathbf{e}^t$ caused by clipping operation gives an increasing factor of $T$ depending on the gradient.

3. By substituting the bound of $\mathbf{e}^t$ into the convergence result in step 1, and choosing sufficiently small stepsize that compensates for the increasing dependency on $T$, we are able to derive a non-trivial convergence result for DiceSGD. A similar result can also be derived for problems satisfying Assumption 3.4.

Our analysis does not apply to the more general non-convex problem, because bounding the error term $\mathbb{E}\left\|\mathbf{e}^t\right\|^2$ relies crucially on the strong convexity assumption. The detailed proof of the algorithm is given in Appendix A.1. A similar result also holds when replacing Assumption 3.3 with Assumption 3.4, and we provide the corresponding theorem and its proof in Appendix A.

Theorem 3.7 indicates that we can choose

$$\eta = \min\left\{\frac{3}{32L}, \frac{\sqrt{C_2}}{5L\sqrt{C_2 + T\max\{0, G' - C_2\}}}\}, \frac{\log\left(2\mu^2 T\mathcal{L}^0 / L(d\sigma_1^2 + \frac{5\sigma^2}{4B} + \frac{C_1^2}{2})\right)}{\mu T}\right\}.$$

When $T$ is sufficiently large, $\eta = \mathcal{O}(\frac{\log(T)}{\mu T})$, so the overall convergence rate for DiceSGD is $\mathcal{O}\left(1 - \frac{\log(T)}{T})^T\right) + \mathcal{O}\left(\frac{\log(T)}{T}\right) = \mathcal{O}\left(\frac{\log(T)}{T}\right).$

Compared with the $\mathcal{O}(\frac{1}{T})$ lower bound convergence rate of DPSGD without gradient clipping under strong convexity (Bassily et al., 2014; Rakhlin et al., 2011), DiceSGD slows down by a factor of $\mathcal{O}(\log(T))$. However, compared with DPSGD-GC (Koloskova et al., 2023), DiceSGD fully eliminates the constant bias, and improves the convergence rate from $\mathcal{O}(1)$ to $\mathcal{O}(\frac{\log(T)}{T})$. The comparison is shown in Table 1.

**Privacy guarantee** Let us proceed with the privacy analysis of DiceSGD. We start with the notion of Rényi Differential Privacy (Mironov, 2017). By accounting for the distribution divergence of the stochastic gradient at iteration $t$ and the accumulated difference of $\mathbf{e}^t$ starting from $\mathbf{e}^0$, we are able to bound the Rényi divergence of $\mathbf{x}^{t+1}$ given two adjacent datasets $\mathcal{D}, \mathcal{D}'$ and start with the same $\mathbf{x}^t$. Then by using the composition theorem of Rényi divergence, we provide the privacy guarantee for DiceSGD in the next result. Note that this result is independent of problem types as we do not assume either Assumption 3.3 and Assumption 3.4.

**Theorem 3.8.** *Assume the problem satisfies Assumptions 3.5, and 3.6, given constant $C$, by fixing the clipping thresholds $0 < C_1 \leq C_2 \leq C/B$, independent of $G, \sigma$, and assume $\frac{B}{N} \leq \frac{1}{5}$. Choose DP noise standard deviation $\sigma_1$ as*

$$\sigma_1^2 \geq \frac{32T\tilde{G}\log(1/\delta)}{N^2\epsilon^2},$$

*where $\tilde{G} := C_1^2 + 2\min\{C^2, G'^2\}$, and $G'$ defined in Theorem 3.7. Running DiceSGD for $T$ iteration, the algorithm guarantees $(\epsilon, \delta)$-differentially private.*

Note that although Assumptions 3.5, and 3.6 are used in the proof, the result does not rely on the specific values of the bounds, which can be arbitrarily large. Due to the accumulated influence of the

Table 2: Test accuracy of DPSGD-GC and DiceSGD on Cifar-10 and Cifar-100 datasets with different clipping thresholds.

| Dataset | Clipping. | DPSGD-GC | DiceSGD | SGD |
|---|---|---|---|---|
| Cifar-10 | $C = 1.0$ | 95.2% | 97.4% | 99.0% |
| Cifar-10 | $C = 0.1$ | 94.5% | 97.5% | 99.0% |
| Cifar-100 | $C = 1.0$ | 79.0% | 86.3% | 92.0% |
| Cifar-100 | $C = 0.1$ | 78.9% | 86.5% | 92.0% |

update of $\mathbf{e}^t$, the DiceSGD requires larger DP-noise than the DPSGD algorithm (larger by a constant multiplicative factor). The detailed proof is given in Appendix A.2. By optimizing $T$ we have the following utility-privacy trade-off for DiceSGD.

**Corollary 3.9.** *Under the same assumptions of Theorem 3.7, choose the stepsize $\eta = \mathcal{O}(T^{-(1-\nu)})$, where $0 < \nu \leq \frac{1}{2}$ is a small constant, and clipping thresholds $0 < C_1 \leq C_2 \leq C/B$, and choose noise multiplier $\sigma_1^2$ as Theorem 3.8. By running DiceSGD for $T = \mathcal{O}\left(\frac{N^2\epsilon^2}{\tilde{G}\log(1/\delta)}\right)$ iterations, the algorithm guarantees $(\epsilon, \delta)$-DP, while converging to a solution where the loss function satisfies:*

$$\mathbb{E}[\mathcal{L}^{T+1}] = \mathcal{O}\left(\frac{\tilde{G}\log(N\epsilon)\log(1/\delta)}{N^2\epsilon^2}\right).$$

A similar result can also be derived by replacing Assumption 3.3 with Assumption 3.4.

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

**Lemma A.2** (Theorem 2 Karimi et al. (2016a)). *Assume a function $f$ satisfies Assumptions 3.2 and 3.3 with constants $L$ and $\mu$, respectively, then we have $f$ satisfies Assumption 3.4 with parameter $\mu$.*

We provide the statements of Theorem 3.7 under the PL condition assumption 3.4 and Theorem 3.8 below for convenience.

**Theorem A.3.** *Assume the problem satisfies Assumption 3.2, 3.4, 3.5, and 3.6. Given any constant DP noise multiplier $\sigma_1$, by running DiceSGD (Algorithm 2) for $T$ iterations, choosing stepsize $\eta = \min\{\frac{1}{2\mu}, \frac{3}{32L}, \frac{\sqrt{C_2}}{5L\sqrt{C_2 + T\max\{0, G' - C_2\}}}\}$, $G' = \max\{0, G + \sigma - C_1\}$, clipping thresholds $C_2 \geq C_1 > 0$. It satisfies*

$$\mathbb{E}[\mathcal{L}^{T+1}] \leq (1 - \mu\eta)^T \mathcal{L}^0 + \frac{\eta L}{2\mu}\left(d\sigma_1^2 + \frac{5\sigma^2}{4B} + \frac{C_1^2}{2}\right) \tag{7}$$

*where $\mathcal{L}^t = f(\tilde{\mathbf{x}}^t) - f^\star + \frac{\eta^3 L^2}{6}\|\mathbf{e}^t\|^2 + \frac{\eta(1 - L\eta)}{2}\|\nabla f(\mathbf{x}^t)\|^2$.*

We see that Theorem 3.7 can be derived from Theorem A.3 and Lemma A.2, because whenever we use Assumption 3.4 for proving Theorem A.3, we can apply Lemma A.2 and use Assumption 3.3 instead. Therefore, we provide the proof for Theorem A.3 in Appendix A.1.

**Theorem A.4.** *Under Assumptions 3.5 and 3.6, given constant $C$, choose the clipping thresholds $0 < C_1 \leq C_2 \leq C/B$. Choosing DP noise multiplier $\sigma_1$ as*

$$\sigma_1^2 \geq \frac{32T(C_1^2 + 2\min\{C^2, G'^2\})\log(1/\delta)}{N^2\epsilon^2}.$$

*Running DiceSGD for $T$ iteration, the algorithm guarantees $(\epsilon, \delta)$-DP.*

### A.1 PROOF OF THEOREM 3.7

Before we provide the convergence proof of Theorem 3.7, let us first define several additional notations which simplify the analysis. We first define a *virtual* update sequence

$$\tilde{\mathbf{x}}^t = \mathbf{x}^t - \eta \mathbf{e}^t, \tag{8}$$

From the update rules in Algorithm 2, we can obtain the update rule for the virtual sequence as

$$
\begin{aligned}
\tilde{\mathbf{x}}^{t+1} &= \mathbf{x}^{t+1} - \eta \mathbf{e}^{t+1} \\
&\overset{(a)}{=} \mathbf{x}^t - \eta \mathbf{v}^t - \eta \mathbf{w}^t - \eta \mathbf{e}^{t+1} \\
&\overset{(b)}{=} \mathbf{x}^t - \eta \mathbf{v}^t - \eta \mathbf{w}^t - \eta(\mathbf{e}^t + \frac{1}{B}\sum_{i\in\mathcal{B}^t}\mathbf{g}_i^t - \mathbf{v}^t) \\
&= \mathbf{x}^t - \eta \mathbf{e}^t - \frac{\eta}{B}\sum_{i\in\mathcal{B}^t}\mathbf{g}_i^t - \eta \mathbf{w}^t \\
&= \tilde{\mathbf{x}}^t - \frac{\eta}{B}\sum_{i\in\mathcal{B}^t}\mathbf{g}_i^t - \eta \mathbf{w}^t,
\end{aligned} \tag{9}
$$

where $(a)$ applies the update rule of $\mathbf{x}^t$ (Algorithm 2 Line 5-6); $(b)$ applies the update rule of $\mathbf{e}^t$ (Algorithm 2 Line 7). Notice that the virtual sequence has the same update rule of DPSGD *without* the clipping operation.

Additionally, let us denote the clipping factors as $\alpha_i^t := \min\left\{1, \frac{C_1}{\|\mathbf{g}_i^t\|}\right\}$, $\alpha_{\mathbf{e}}^t := \min\left\{1, \frac{C_2}{\|\mathbf{e}^t\|}\right\}$, so that

$$\text{clip}\left(\mathbf{e}^t, C_2\right) = \alpha_{\mathbf{e}}^t \mathbf{e}^t,\ \text{clip}\left(\mathbf{g}_i^t, C_1\right) = \alpha_i^t \mathbf{g}_i^t,\ \text{and}\ \mathbf{e}^{t+1} = (1-\alpha_{\mathbf{e}}^t)\mathbf{e}^t + \frac{1}{B}\sum_{i\in\mathcal{B}^t}(1-\alpha_i^t)\mathbf{g}_i^t. \tag{10}$$

First, with Assumption 3.2 and let $\mathbb{E}_t$ denote the conditional expectation given history information from iteration 0 up to $t-1$ and condition on the randomness at iteration $t$, then we have:

$$
\mathbb{E}_t[f(\tilde{\mathbf{x}}^{t+1}) - f^\star] \le f(\tilde{\mathbf{x}}^t) - f^\star + \mathbb{E}_t\left\langle \nabla f(\tilde{\mathbf{x}}^t), \tilde{\mathbf{x}}^{t+1} - \tilde{\mathbf{x}}^t \right\rangle + \mathbb{E}_t\left[\frac{L}{2}\left\|\tilde{\mathbf{x}}^{t+1} - \tilde{\mathbf{x}}^t\right\|^2\right]
$$

$$
\overset{(9)}{=} f(\tilde{\mathbf{x}}^t) - f^\star + \frac{L\eta^2}{2}\mathbb{E}_t\left[\left\|\frac{1}{B}\sum_{i\in\mathcal{B}^t}\mathbf{g}_i^t - \mathbf{w}^t\right\|^2\right] - \eta\left\langle \nabla f(\tilde{\mathbf{x}}^t), \mathbb{E}_t\left[\frac{1}{B}\sum_{i\in\mathcal{B}^t}\mathbf{g}_i^t - \mathbf{w}^t\right]\right\rangle
$$

$$
\overset{(a)}{\le} f(\tilde{\mathbf{x}}^t) - f^\star - \eta\left\langle \nabla f(\tilde{\mathbf{x}}^t), \nabla f(\mathbf{x}^t)\right\rangle + \frac{L\eta^2}{2}\left(\left\|\nabla f(\mathbf{x}^t)\right\|^2 + \frac{\sigma^2}{B} + d\sigma_1^2\right)
$$

$$
\overset{(6)}{=} f(\tilde{\mathbf{x}}^t) - f^\star - \frac{\eta}{2}\left(\left\|\nabla f(\tilde{\mathbf{x}}^t)\right\|^2 + \left\|\nabla f(\mathbf{x}^t)\right\|^2\right) + \frac{\eta}{2}\left\|\nabla f(\tilde{\mathbf{x}}^t) - \nabla f(\mathbf{x}^t)\right\|^2
$$
$$
+ \frac{L\eta^2}{2}\left(\left\|\nabla f(\mathbf{x}^t)\right\|^2 + \frac{\sigma^2}{B} + d\sigma_1^2\right)
$$

$$
\overset{(b)}{\le} f(\tilde{\mathbf{x}}^t) - f^\star - \frac{\eta}{2}\left\|\nabla f(\tilde{\mathbf{x}}^t)\right\|^2 - \frac{\eta(1-L\eta)}{2}\left\|\nabla f(\mathbf{x}^t)\right\|^2 + \frac{\eta L^2}{2}\left\|\tilde{\mathbf{x}}^t - \mathbf{x}^t\right\|^2
$$
$$
+ \frac{L\eta^2}{2}\left(\frac{\sigma^2}{B} + d\sigma_1^2\right)
$$

$$
\overset{(9)}{=} f(\tilde{\mathbf{x}}^t) - f^\star - \frac{\eta}{2}\left\|\nabla f(\tilde{\mathbf{x}}^t)\right\|^2 - \frac{\eta(1-L\eta)}{2}\left\|\nabla f(\mathbf{x}^t)\right\|^2 + \frac{\eta^3 L^2}{2}\left\|\mathbf{e}^t\right\|^2 + \frac{L\eta^2}{2}\left(\frac{\sigma^2}{B} + d\sigma_1^2\right)
$$

$$
\overset{(c)}{\le} (1-\eta\mu)[f(\tilde{\mathbf{x}}^t) - f^\star] - \frac{\eta(1-L\eta)}{2}\left\|\nabla f(\mathbf{x}^t)\right\|^2 + \frac{\eta^3 L^2}{2}\left\|\mathbf{e}^t\right\|^2 + \frac{L\eta^2}{2}\left(\frac{\sigma^2}{B} + d\sigma_1^2\right), \tag{11}
$$

where $(a)$ applies Assumption 3.5 and use the fact that $\mathbf{w}^t$ is an independent Gaussian noise from the stochastic gradient; $(b)$ applies Assumption 3.2 to the third term, and $(c)$ applies Assumption 3.4.

Next, we bound $\left\|\mathbf{e}^t\right\|^2$ from its update rule:

$$\mathbb{E}_{t-1}\left\|\mathbf{e}^t\right\|^2 \overset{(10)}{=} \mathbb{E}_{t-1}\left\|(1-\alpha_{\mathbf{e}}^{t-1})\mathbf{e}^{t-1} + \frac{1}{B}\sum_{i\in\mathcal{B}^{t-1}}(1-\alpha_i^{t-1})\mathbf{g}_i^{t-1}\right\|^2$$

$$\overset{(5)}{\leq} (1+\beta^{t-1})(1-\alpha_{\mathbf{e}}^{t-1})^2\left\|\mathbf{e}^{t-1}\right\|^2 + (1+\frac{1}{\beta^{t-1}})\mathbb{E}_{t-1}\left\|\frac{1}{B}\sum_{i\in\mathcal{B}^{t-1}}(1-\alpha_i^{t-1})\mathbf{g}_i^{t-1}\right\|^2$$

$$\overset{(a)}{=} (1+\beta^{t-1})(1-\alpha_{\mathbf{e}}^{t-1})^2\left\|\mathbf{e}^{t-1}\right\|^2 + (1+\frac{1}{\beta^{t-1}})\left\|\mathbb{E}_{t-1}[(1-\alpha_i^{t-1})\mathbf{g}_i^{t-1}]\right\|^2$$

$$+ (1+\frac{1}{\beta^{t-1}})\mathbb{E}_{t-1}\left\|\frac{1}{B}\sum_{i\in\mathcal{B}^{t-1}}((1-\alpha_i^{t-1})\mathbf{g}_i^{t-1} - \mathbb{E}_{t-1}[(1-\alpha_i^{t-1})\mathbf{g}_i^{t-1}])\right\|^2$$

$$\overset{(b)}{=} (1+\beta^{t-1})(1-\alpha_{\mathbf{e}}^{t-1})^2\left\|\mathbf{e}^{t-1}\right\|^2 + (1+\frac{1}{\beta^{t-1}})\left\|\mathbb{E}_{t-1}[(1-\alpha_i^{t-1})\mathbf{g}_i^{t-1}]\right\|^2$$

$$+ (1+\frac{1}{\beta^{t-1}})\frac{1}{B^2}\sum_{i\in\mathcal{B}^{t-1}}\mathbb{E}_{t-1}\left\|(1-\alpha_i^{t-1})\mathbf{g}_i^{t-1} - \mathbb{E}_{t-1}[(1-\alpha_i^{t-1})\mathbf{g}_i^{t-1}]\right\|^2$$

$$+ (1+\frac{1}{\beta^{t-1}})\frac{1}{B^2}\sum_{i\neq j\in\mathcal{B}^{t-1}}\mathbb{E}_{t-1}\left\langle(1-\alpha_i^{t-1})\mathbf{g}_i^{t-1} - \mathbb{E}_{t-1}[(1-\alpha_i^{t-1})\mathbf{g}_i^{t-1}], (1-\alpha_j^{t-1})\mathbf{g}_j^{t-1} - \mathbb{E}_{t-1}[(1-\alpha_i^{t-1})\mathbf{g}_i^{t-1}]\right\rangle$$

$$\overset{(c)}{\leq} (1+\beta^{t-1})(1-\alpha_{\mathbf{e}}^{t-1})^2\left\|\mathbf{e}^{t-1}\right\|^2 + (1+\frac{1}{\beta^{t-1}})\left\|\mathbb{E}_{t-1}\,\mathbf{g}_i^{t-1} - \mathbb{E}_{t-1}\,\alpha_i^{t-1}\mathbf{g}_i^{t-1}\right\|^2 + (1+\frac{1}{\beta^{t-1}})\frac{\sigma^2}{B}$$

$$\overset{(5)}{\leq} (1+\beta^{t-1})(1-\alpha_{\mathbf{e}}^{t-1})^2\left\|\mathbf{e}^t\right\|^2 + 2(1+\frac{1}{\beta^{t-1}})\left(\left\|\mathbb{E}_{t-1}\,\mathbf{g}_i^{t-1}\right\|^2 + \left\|\mathbb{E}_{t-1}\,\alpha_i^{t-1}\mathbf{g}_i^{t-1}\right\|^2\right) + (1+\frac{1}{\beta^{t-1}})\frac{\sigma^2}{B},$$

$$\overset{(d)}{\leq} (1+\beta^{t-1})(1-\alpha_{\mathbf{e}}^{t-1})^2\left\|\mathbf{e}^t\right\|^2 + 2(1+\frac{1}{\beta^{t-1}})\left(\left\|\nabla f(\mathbf{x}^{t-1})\right\|^2 + C_1^2\right) + (1+\frac{1}{\beta^{t-1}})\frac{\sigma^2}{B}, \tag{12}$$

where $(a)$ uses the fact that $\mathbb{E}[X]^2 = [\mathbb{E}\,X]^2 + \text{Var}(X)$; $(b)$ expands the last term of $(a)$; $(c)$ applies Lemma A.1 to the second last term and applies the fact that samples in $\mathcal{B}$ are independently sampled from $\mathcal{D}$ so the last term equals to 0; and $(d)$ applies Assumption 3.6 and notice the last term is the clipped gradient so $\left\|\mathbb{E}_{t-1}\,\alpha_i^{t-1}\mathbf{g}_i^{t-1}\right\|^2 \leq C_1^2$. Adding $\frac{\gamma\eta^3 L^2}{2}\times$(12) to (11), we have:

$$\mathbb{E}_t\left[f(\tilde{\mathbf{x}}^{t+1}) - f^\star + \frac{(1-\gamma)\eta^3 L^2}{2}\left\|\mathbf{e}^t\right\|^2 + \frac{\eta(1-L\eta)}{2}\left\|\nabla f(\mathbf{x}^t)\right\|^2\right] \leq \mathbb{E}_t\left[(1-\eta\mu)f(\tilde{\mathbf{x}}^t) - f^\star\right]$$

$$+ \mathbb{E}_t\left[2\gamma\eta^3 L^2(1+\frac{1}{\beta^{t-1}}))\left\|\nabla f(\mathbf{x}^{t-1})\right\|^2 + \frac{\gamma\eta^3 L^2(1+\beta^{t-1})(1-\alpha_{\mathbf{e}}^{t-1})^2}{2}\left\|\mathbf{e}^{t-1}\right\|^2\right]$$

$$+ \frac{L\eta^2}{2}\left(\frac{(1+\gamma\eta L(1+\frac{1}{\beta^{t-1}}))\sigma^2}{B} + d\sigma_1^2 + 2\gamma\eta L(1+\frac{1}{\beta^{t-1}})C_1^2\right), \tag{13}$$

Setting the stepsize $\eta$, and constants $\gamma$, $\beta^{t-1}$ as follows

$$0 < \eta \leq \min\left\{\frac{1}{\mu}, \frac{\alpha_{\mathbf{e}}^{t-1}\left(\sqrt{1+16\gamma\frac{1+\alpha_{\mathbf{e}}^{t-1}}{\alpha_{\mathbf{e}}^{t-1}}}-1\right)}{8\gamma(1+\alpha_{\mathbf{e}}^{t-1})L}\right\}, \quad \gamma \leq \frac{1}{2+\alpha_{\mathbf{e}}^{t-1}}, \quad \beta^{t-1} = \alpha_{\mathbf{e}}^{t-1}, \tag{14}$$

we can guarantee that the following relations hold:

$$1 - L\eta \geq 4\gamma\eta^2 L^2(1+\frac{1}{\beta^{t-1}}), \; 1 \geq \eta\mu > 0, \text{ and } 1-\gamma \geq \gamma(1+\beta^{t-1})(1-\alpha_{\mathbf{e}}^{t-1})^2.$$

Let us define the following quantities:

$$\mathcal{L}^t := f(\tilde{\mathbf{x}}^t) - f^\star + \frac{\gamma\eta^3 L^2}{2}\left\|\mathbf{e}^t\right\|^2 + \frac{\eta(1-L\eta)}{2}\left\|\nabla f(\mathbf{x}^t)\right\|^2, \tag{15a}$$

$$\text{and} \quad \kappa := \min\left\{\eta\mu, 1 - \frac{\gamma(1+\beta^{t-1})(1-\alpha_{\mathbf{e}}^t)^2}{1-\gamma}, 1 - \frac{4\gamma\eta^2 L^2(1+\frac{1}{\beta^{t-1}})}{1-L\eta}\right\} \tag{15b}$$

Then we can simplify (13) to

$$\mathbb{E}_t[\mathcal{L}^{t+1}] \le (1-\kappa)\mathcal{L}^t + \frac{L\eta^2}{2}\left(\frac{(1+\gamma\eta L(1+\frac{1}{\beta^{t-1}}))\sigma^2}{B} + d\sigma_1^2 + 2\gamma\eta L(1+\frac{1}{\beta^{t-1}})C_1^2\right)$$

$$= (1-\kappa)\mathcal{L}^t + \frac{L\eta^2}{2}\Delta^t, \tag{16}$$

where we have defined $\Delta^t := \frac{(1+\gamma\eta L(1+\frac{1}{\beta^{t-1}}))\sigma^2}{B} + d\sigma_1^2 + 2\gamma\eta L(1+\frac{1}{\beta^{t-1}})C_1^2$. Therefore, by taking the full expectation, we have:

$$\mathbb{E}[\mathcal{L}^{t+1}] \le (1-\kappa)\mathbb{E}[\mathcal{L}^t] + \frac{\eta^2 L}{2}\Delta^t$$

$$\overset{(a)}{\le} (1-\kappa)^t\mathcal{L}^0 + \frac{\eta^2 L}{2}\sum_{\tau=0}^{t}(1-\kappa)^{t-\tau}\Delta^\tau \tag{17}$$

$$\overset{(b)}{\le} (1-\kappa)^t\mathcal{L}^0 + \frac{\eta^2 L}{2\kappa}\max_\tau\{\Delta^\tau\},$$

where $(a)$ recursively expands $\mathbb{E}[\mathcal{L}^t]$ to $\mathcal{L}^0$, and $(b)$ applies $\sum_{\tau=0}^{t}(1-\kappa)^\tau = \frac{1-(1-\kappa)^{t+1}}{\kappa} \le \frac{1}{\kappa}$.

In order to derive the final convergence result, we need to determine $\kappa, \eta$, and $\Delta^t$, which all depend on $\alpha_{\mathbf{e}}^t$.

Next, we provide a bound for $\alpha_{\mathbf{e}}^t$. Let us consider two cases: 1) $\|\mathbf{e}^t\| \le C_2$ and $\|\mathbf{e}^t\| > C_2$. For case 1), it is clear that $\alpha_{\mathbf{e}}^t = 1$. For case 2), we have $\alpha_{\mathbf{e}}^t = \frac{C_2}{\|\mathbf{e}^t\|}$. We can further bound $\|\mathbf{e}^t\|$ recursively as

$$\|\mathbf{e}^t\| \overset{(10)}{=} \left\|(1-\alpha_{\mathbf{e}}^{t-1})\mathbf{e}^{t-1} + \frac{1}{B}\sum_{i\in\mathcal{B}^{t-1}}(1-\alpha_i^{t-1})g_i^{t-1}\right\|$$

$$\le (1-\alpha_{\mathbf{e}}^{t-1})\|\mathbf{e}^{t-1}\| + \left\|\frac{1}{B}\sum_{i\in\mathcal{B}^{t-1}}(1-\alpha_i^{t-1})g_i^{t-1}\right\|$$

$$\overset{(a)}{\le} (1-\alpha_{\mathbf{e}}^{t-1})\|\mathbf{e}^{t-1}\| + \frac{1}{B}\sum_{i\in\mathcal{B}^{t-1}}\left\|\left(1-\min\{1,\frac{C_1}{\|g_i^{t-1}\|}\}\right)g_i^{t-1}\right\|$$

$$\overset{(b)}{\le} (1-\alpha_{\mathbf{e}}^{t-1})\|\mathbf{e}^{t-1}\| + \frac{1}{B}\sum_{i\in\mathcal{B}^{t-1}}\max\{0,\|g_i^{t-1}\| - C_1\}$$

$$\overset{(c)}{\le} \|\mathbf{e}^{t-1}\| - C_2 + \max\{0,\|\nabla f(\mathbf{x}^{t-1})\| + \sigma - C_1\}$$

$$\overset{(d)}{\le} \|\mathbf{e}^1\| + \sum_{\tau=1}^{t-1}(\max\{0,\|\nabla f(\mathbf{x}^\tau)\| + \sigma - C_1\} - C_2)$$

$$\overset{(e)}{\le} t\max\{0, G+\sigma-C_1\} - (t-1)C_2 = tG' - (t-1)C_2.$$

where $(a)$ substitutes the definition of $\alpha_{\mathbf{e}}^t$; $(b)$ expands the last term; $(c)$ applies Assumption 3.5; in $(d)$ we recursively expand $\|\mathbf{e}^{t-1}\|$ to $\|\mathbf{e}^1\|$ and notice that $\|\mathbf{e}^1\| = \|\frac{1}{B}\sum_{i\in\mathcal{B}^1}(1-\alpha_i^1)g_i^1\| \le \max\{0,\|\nabla f(\mathbf{x}^1)\| + \sigma - C_1\}$; in $(e)$ we apply Assumption 3.6 and define $G' := \max\{0, G+\sigma-C_1\}$. Therefore, we have

$$\alpha_{\mathbf{e}}^t = \frac{C_2}{\|\mathbf{e}^t\|} \ge \frac{C_2}{tG' - (t-1)C_2} = \frac{C_2}{C_2 + t(G'-C_2)}. \tag{18}$$

By substituting the above lower bound of $\alpha_{\mathbf{e}}^t$ back into the upper bounds of $\eta, \beta^t, \gamma$ in (14), we will get explicit bounds for these constants and $\kappa, \Delta^t$. First, it is clear that we can set $\gamma = \frac{1}{3} \le \frac{1}{2+\alpha_{\mathbf{e}}^t}$ because $\alpha_{\mathbf{e}}^t \le 1$.

Next, we derive the range of $\dfrac{\alpha_{\mathbf{e}}^{t-1}\left(\sqrt{1+16\gamma\frac{1+\alpha_{\mathbf{e}}^{t-1}}{\alpha_{\mathbf{e}}^{t-1}}}-1\right)}{8\gamma(1+\alpha_{\mathbf{e}}^{t-1})L}$ in the upper bound of $\eta$ that $\eta \leq$

$\min\left\{\dfrac{1}{\mu},\dfrac{\alpha_{\mathbf{e}}^{t-1}\left(\sqrt{1+16\gamma\frac{1+\alpha_{\mathbf{e}}^{t-1}}{\alpha_{\mathbf{e}}^{t-1}}}-1\right)}{8\gamma(1+\alpha_{\mathbf{e}}^{t-1})L}\right\}$ . We analyze it in two cases: 1) $\alpha_{\mathbf{e}}^{t-1} \leq \frac{1}{2}$ and 2) $\alpha_{\mathbf{e}}^{t-1} > \frac{1}{2}$. In

case 1) where $\alpha_{\mathbf{e}}^{t-1} \geq \frac{1}{2}$,

$$\frac{\alpha_{\mathbf{e}}^{t-1}\left(\sqrt{1+16\gamma\frac{1+\alpha_{\mathbf{e}}^{t-1}}{\alpha_{\mathbf{e}}^{t-1}}}-1\right)}{8\gamma(1+\alpha_{\mathbf{e}}^{t-1})L} \geq \frac{3}{16L},$$

and in case 2) where $\alpha_{\mathbf{e}}^{t-1} < \frac{1}{2}$,

$$\frac{\alpha_{\mathbf{e}}^{t-1}\left(\sqrt{1+16\gamma\frac{1+\alpha_{\mathbf{e}}^{t-1}}{\alpha_{\mathbf{e}}^{t-1}}}-1\right)}{8\gamma(1+\alpha_{\mathbf{e}}^{t-1})L} > \frac{1.6\sqrt{\alpha_{\mathbf{e}}^{t-1}}}{4L} \geq \frac{2\sqrt{C_2}}{5L\sqrt{C_2+t(G'-C_2)}}.$$

Therefore, we can set $\eta = \min\left\{\dfrac{1}{2\mu},\dfrac{3}{32L},\dfrac{\sqrt{C_2}}{5L\sqrt{C_2+t(G'-C_2)}}\right\}$, and from (15b), we obtain $\kappa =$

$\min\{\frac{1}{2},\frac{3\mu}{32L},\frac{\mu\sqrt{C_2}}{5L\sqrt{C_2+t(G'-C_2)}}\} = \eta\mu$. By substituting the above choices of $\eta,\kappa$ and $\beta^t$ into the definition of $\Delta^t$ that

$$\begin{aligned}
\Delta^t &= \frac{(1+\gamma\eta L(1+\frac{1}{\beta^{t-1}}))\sigma^2}{B} + d\sigma_1^2 + 2\gamma\eta L(1+\frac{1}{\beta^{t-1}})C_1^2 \\
&\leq \frac{\sigma^2}{B}(1+\frac{\sqrt{1+\frac{32}{3}}-1}{16}) + d\sigma_1^2 + 2\frac{\sqrt{1+\frac{32}{3}}-1}{16}C_1^2 \\
&\leq \frac{5\sigma^2}{4B} + d\sigma_1^2 + \frac{C_1^2}{2}.
\end{aligned}$$

Substitute $\eta,\kappa$ and $\Delta^t$ into (16), we obtain

$$\mathbb{E}[\mathcal{L}^{T+1}] \leq (1-\mu\eta)^T\mathcal{L}^0 + \frac{\eta L}{2\mu}\left(\frac{5\sigma^2}{4B} + d\sigma_1^2 + \frac{C_1^2}{2}\right).$$