# OpenReview forum: "Differentially Private SGD Without Clipping Bias: An Error-Feedback Approach"
_ICLR.cc/2024/Conference — ICLR 2024 poster_

### Official Review · Reviewer_KeFV · 2023-10-30

**Soundness:** 3 good
**Presentation:** 3 good
**Contribution:** 3 good
**Rating:** 6
**Confidence:** 4

**Summary:**

A novel algorithm for stochastic differentially private model training is proposed that eliminates the bias caused by clipping. The essential idea is to maintain the clipped component in a separate accumulator that is also added (with its own clipping) to the update at each iteration. A convergence proof is provided (for strongly convex loss) showing that the algorithm does not suffer from the O(1) term caused by clipping in typical DP-SGD. Experimental results show that the method closes the gap to non-private SGD on several tasks.

**Strengths:**

The algorithm is novel and, while not terribly surprising in hindsight (like many good ideas) is a useful contribution. More original is the DP analysis, which is complicated due to the non-privatized hidden clipping-error-accumulator state. The experimental results (on non-convex objectives) are encouraging, and it is in particular very exciting that the dependence on the clipping threshold appears to be weak. If future results continue to show such a pattern, that would be particularly advantagous, as DP-SGD can be quite sensitive to the clipping threshold.

**Weaknesses:**

It is unfortunate that the convergence result depends on strong convexity. This limits the usefulness of the result. Fortunately the DP guarantee does not require the strong convexity assumption, and the experimental results use non-convex models.

The experiments don't seem to be very careful about how $C$ is chosen. For example, in Table 3, it is just stated that C=1. Considering the whole point of the paper is a better method for clipping, I would like to see more exploration into this. In Table 2, we just look at two points C=1.0 and C=0.1 for each dataset. But there could be an interesting curve there because there is a tradeoff between clipping error (which is hopefully alleviated by DiceSGD) and the amount of noise you have to add for DP.

**Questions:**

* It is stated that DiceSGD requires a constant multiplicative factor more noise than DPSGD. What is that factor?
* Given that DiceSGD maintains a private state anyway, and the convergence analysis assumes strong convexity, could the privacy guarantee be strengthened by NOT releasing the model after every step, similar to Feldman et al. "Privacy Amplification by Iteration" (which is cited)? The analysis you already have -- for non-convex objectives and releasing all iterates -- seems to me primarily useful because it bounds the privacy loss for non-convex objectives *without* releasing all iterates. I can't imagine a scenario where you can protect the private state but nevertheless want to release all model iterates. On the other hand, if it were possible, an analysis of the convex case where all model iterates are private could give you a tighter bound.
* In the NLP experiment, it is stated that GPT-2 is fine-tuned. By default I would assume that means all-parameter fine-tuning, but Hu et al. (2021) is cited twice (referencing the metrics and in the table) which makes me wonder if it is Lora fine tuning. Which is it?

---

> ### Author Response · Authors · 2023-11-23
> **Response to Reviewer KeFV**
>
> ### How $C$ is chosen:
> > We provide additional experiments studying the affect of the clipping threshold. Please refer to additional experiments in the appendix (figure 1, 2 and 4). Figure 1 and 2 study how the choice of $C_1, C_2$ and $\eta$ affects the final accuracy of the CV task. Figure 4 also studies different choices of $C_1, \eta$ on the NLP task.
>
> ### Constant noise factor for DiceSGD
> > If DiceSGD and DPSGD are using the same clipping factor $C$, then the factor is $8$; However, DiceSGD allows us to choose a smaller clipping factor without affecting the accuracy.
>
> ### On privacy amplification by iteration
> > Indeed, that we think privacy amplification by iteration can be used (Ye & Shokri, 2022, as mentioned in Sec. 3.2). This would be a future work. However, in current stage, we are still comparing with DPSGD, which releases all intermediate model states.
>
> ### On fine-tuning method
> >Let us clarify that we are using full model fine-tuning instead of LoRA.

---

> > ### Comment · Reviewer_KeFV · 2023-11-23
> >
> > Thank you for your response.

---

### Official Review · Reviewer_w9rc · 2023-11-01

**Soundness:** 3 good
**Presentation:** 3 good
**Contribution:** 2 fair
**Rating:** 6
**Confidence:** 2

**Summary:**

This paper proposes a new variant of DP-SGD algorithm that eliminates the clipping bias during training. The key idea is to borrow the error-feedback mechanism for compressed SGD. Theoretically, under some common assumptions, this paper provides novel convergence analysis and privacy analysis. Finally, the effectiveness of the proposed algorithm is demonstrated on empirical experiments.

**Strengths:**

1. This paper is well-written.
2. The proposed algorithm is relatively simple and scalable. The algorithm itself does not require fine-tuning the clipping threshold.
3. Proofs of utility and privacy both require non-trivial effort and novel technique.

**Weaknesses:**

Compared to DP-SGD with a properly chosen clipping threshold, the error rate is larger, and the convergence is slower. This is due to the larger noise needs to be added. I was wondering if there is any practical implication in the experiments.

**Questions:**

Please see weaknesses

---

### Official Review · Reviewer_uaba · 2023-11-01

**Soundness:** 3 good
**Presentation:** 3 good
**Contribution:** 3 good
**Rating:** 6
**Confidence:** 3

**Summary:**

The paper to add a state accumulating the error caused by clipping in DP-SGD and adding it back later, which is called DiceSGD. It is designed to reduce the clipping bias of DP-SGD

**Strengths:**

1. This is an interesting attempt at analyzing and mitigating the clipping bias in DP-SGD.
2. The convergence analysis and privacy analysis are non-trivial.

**Weaknesses:**

1. My major concern is that although clipping bias can cause privacy degradation in DP-SGD, the major drop is still caused by the added noise. Thus I am not sure how useful it is to trade noise variance for clipping bias. The empirical study partially answers the question but might not be generalizable.
2. I think DiceSGD should be compared with DP-FTRL and matric mechanism as well since they are all stateful algorithms. In terms of implementation, DP-SGD is more efficient than all of them since it is stateless.

**Questions:**

Is it possible to get a rigorous bias-variance trade-off to confirm that DiceSGD is better than DP-SGD?

---

### Official Review · Reviewer_abPz · 2023-11-02

**Soundness:** 3 good
**Presentation:** 2 fair
**Contribution:** 3 good
**Rating:** 6
**Confidence:** 3

**Summary:**

The paper proposes a differentially private optimization algorithm with error feedback as a means to combat the clipping bias. The resulting algorithm is shown to converge without a bias at all values of the clip norm (whereas DP-SGD analyses show convergence at particular clip norm values). Experiments show that the proposed approach outperforms DP-SGD. The theoretical analysis requires a strong assumption that amounts to requiring that all per-example gradients are almost surely bounded.

**Strengths:**

- The result is quite significant. Clipping is a known problem in the literature both in theory (where existing analyses are inadequate) and in practice (where the clip norm needs to be tuned carefully). This paper addresses this important problem.
- The main paper is reasonably well-written and clear to understand. The fixed point analysis used to motivate the method does a great job of explaining the key ideas. The proofs could use some rewriting to improve clarity (details below).
- The optimization proofs look right to me, although I have not verified them line-by-line. I was unable to parse the privacy proofs which could use some rewriting.
- I would rate the originality as moderate since the paper is the application of a well-known technique from optimization (and signal processing before that) to DP optimization. This application introduces some technical challenges in the proof which the paper handles well.

**Weaknesses:**

**Theory**:
- _Discussion of assumptions_: The assumption that $\|\nabla f(x) - g_i\|_2 \le \sigma$ everywhere is rather strong. It needs to be better and more transparently justified. Its ramifications need to be discussed more carefully. For instance, under such an assumption, it is not inconceivable that DPSGD with [adaptive clipping](https://arxiv.org/abs/1905.03871) also converges favorably (see e.g. [Varshney et al.](https://arxiv.org/abs/2207.04686) who used this trick for linear regression). A comparison to this approach both theoretically and empirically is missing.
- _Convergence rates_: I believe that $\eta = O(T^{-(1-\nu)})$ is a suboptimal choice and the right rate should be something like $\exp(-c\sqrt{t}) + \log t /t$. See e.g. [Section 3 of Stich](https://arxiv.org/pdf/1907.04232.pdf) for notes on tuning the step size carefully to get the best rates.
- _Setting of parameters: How does the convergence rate vary with $C_1$ and $C_2$? What are their optimal values? How much does the rate suffer when these constants differ from their optimal values?
- _Potentially incorrect statements_: Page 8 says "the result does not rely on the specific values of [$\sigma$ and $G$]". I do not think this is true: $\sigma_1$ depends on $\tilde G$, which depends on $G'$ which depends on $G$ and $\sigma$.
- _Effect of the clip norm on the learning rate_: Suppose the clip norm is so small that all gradients are effectively clipped. Then, reducing the clip norm any further has the role of reducing the learning rate. Do the bounds capture this dependence? More generally, how does the clip norm determine the optimal learning rate?

**Inadequate experiments**:
- The experimental results are promising but many more experiments are needed to paint a convincing picture. It is fine to run some detailed experiments on smaller-scale settings that can be tested more extensively (I would argue that these have more value for this particular paper than inadequate experiments on larger models but the authors can feel free to disagree).
- Standard deviations across multiple repetitions need to be reported as the algorithms are noisy
- How is the clip norm of DPSGD tuned? That is, is the optimal choice one of $C \in \{0.1, 1\}$. If the claim is that DiceSGD converges regardless of the clip norm, one needs to show the results for a broad range of clip norms (including the best possible for DPSGD) ranging over several orders of magnitude.
- How do these results translate to different values of $\epsilon$?

**Clarity (main paper)**:
- The distinction between "DPSGD" (without clipping) and "DPSGD-GC" (with clipping) is confusing. DPSGD is always presented in the literature with clipping and theoretical analyses use Lipschitz functions (=> bounded gradients) to avoid the technicalities induced by clipping. It would be much more natural to refer to both as DPSGD and make a distinction in the setting (i.e., whether the clipping is active or not).
- Section 2.1: why even mention $(\epsilon, \delta)$-DP? It would be much cleaner to describe the background in terms of RDP and the proofs use this as well.
- $\mathbb{E}_i$ and $\mathbf{g}_i$ are not defined but are used in various places (.e.g Assumption 3.5). This is also poor notation because it implicitly hides the dependence on $\mathbf{x}$.

**Clarity (proofs)**:
- The definitions of key quantities such as $\Delta^t$ and $\alpha^t_e$ should be featured more prominently. Right now, they are hidden somewhere amid the proof and are hard to find. E.g. bottom of p.18, $\alpha^t_e$ is used but it is defined in the proof of Theorem 3.7.
- Lemma A.6: $\mathcal{A}_1^t, \mathcal{A}_2^t, \mathcal{H}^t$: I do not understand this notation. Some examples will be helpful.
- p.18, Step II: This proof is very hard to parse and the quantities do not make sense to me. Please consider standardizing notation across various proofs for the reader's convenience and simplifying the notation in this proof in particular.

**Missing refs**:
- Using normalized gradients instead of clipping: https://arxiv.org/pdf/2106.07094.pdf
- Section 2.3: there is some literature on joint DP + compression e.g. https://arxiv.org/abs/2111.00092

**Minor**:
- Why is $\tilde g_i^t$ not boldfaced in Algorithm 1?
- There is $\mathbb{E}$ missing in the "e" terms in the first display equation of page 6.
- Assumption 3.1: typo $\mathbb{R} -> \mathbb{R}^d$. Btw, Assumption 3.1 always holds under strong convexity (which is assumed), so it does not have to be explicitly specified
- Assumption 3.2: reads like a definition more than an assumption
- Theorem 3.7: $\frac{1}{2\mu} \ge \frac{3}{32 L}$ always, so that condition on $\eta$ can be dropped.
- Theorem 3.7: $\kappa$ is typically used to refer to the condition number and can be quite misleading here. Besides, there is no need to define a new variable if it is used only once.
- $\nu$ and $\tilde G$ are not defined in Table 1
- Theorem 3.8: $\sigma_1$ is not a noise multiplier, it is the noise standard deviation. See [here](https://arxiv.org/pdf/2303.00654.pdf) for details on the terminology.
- Formating math: proper-sized brackets (using \left, \right), etc.
- Some references need to be updated (citing arxiv versions instead of the published ones, etc.)
- Typo on p.18, step 1: $\sigma -> \sigma_1$
-Typos in Lemma A.11: do you need $\sigma > 4$ and $\epsilon = 2p^2 \alpha / \sigma^2$.

**Questions:**

I've given several questions in the "weaknesses" section. Some more questions:
- p. 18, Step III: In the definition of $\Delta^t_g$, are the $g_i^t$ under both $D$ and $D'$ assumed to be the same? If the differing example was sampled in iteration $k < t$, then the iterates $\mathbf{x^t}, \mathbf{x'^t}$ are both distinct. Then, their corresponding gradients in iteration $t$ are also distinct.

Overall, I think this idea is great. If the authors can develop the theory further, run comprehensive experiments, and improve the clarity, I think this paper can be really impactful.

---

> ### Author Response · Authors · 2023-11-23
> **Response to Reviewer abPz (Part I)**
>
> ## Theory
> * Discussion on Assumption 3.5, $\|\nabla f(x) - g_i\|\leq \sigma$. First, this assumption is related to the common bounded variance assumption (i.e., $\mathbb{E}\|\nabla f(x)-g_i\|\leq \sigma^2$). Our assumption differs from the bounded variance assumption by a factor of $1/N$, as discussed in page 7 in our original manuscript. Second, let us compare our result with the adaptive clipping result in Varshney et al., 2022: 1) their analysis only applies to the linear regression model, while our convergence result hold for all functions satisfying PL condition; Therefore, we cannot provide empirical comparision in our experiment training the neural networks.; 2) to achieve convergence, their choice of the clipping threshold depends on $\sigma$, covariance matrix $H$ and the optimal solution $x^\star$ (i.e., $C = O(\sqrt{H}\|x^\star\|+\sigma)$), while these values cannot be obtained before training. In contrast, our clipping thresholds are **independent** of the problem parameters (i.e., $\mu, L, G, \sigma$) and can be arbitrarily chosen. Therefore, compared with adaptive clipping, our algorithm imposes fewer constraints to the problem class and clipping threshold, and our As. 3.5 is also less restrictive than the works using the same assumption.
>
> * Convergence rate: we thank the reviewer for pointing out this issue. Indeed, we can follow the same choice of $\eta$ as Stich, 2019 as
> $$\eta = \min\left\\{\frac{3}{32L}, \frac{\sqrt{C_2}}{5L\sqrt{C_2 + T \max\{0, G'-C_2\}}}, \frac{\log(\mu^2 \mathcal{L}^0T/L(d\sigma_1^2+5\sigma^2/4B+C_1^2/2))}{\mu T}\right\\},$$
> and our optimized convergence rate is $\mathcal{O}\left(\exp(-\frac{\mu T}{L})+\frac{\log(T)L}{\mu^2 T}\right)$. Following this modification, the privacy-utility tradeoff of DiceSGD becomes $\mathcal{O}(\frac{\tilde{G}\log(1/\delta)}{N^2\epsilon^2})$. We have update the results in our revised manuscript.
>
> * Convergence with respect to $C_1, C_2$. Starting with Theorem 3.7, and the choice of $\eta$ above, we have the following relationships: $C_2\geq C_1$, $\eta \leq \frac{\sqrt{C_2}}{5L\sqrt{C_2+ T \max\{0, \max\{0, G+\sigma -C_1\}-C_2\}}}$ and $\mathbb{E}[\mathcal{L}^{T+1}]\leq \frac{\eta L}{2\mu}\cdot \frac{C_1^2}{2}$. From these relations, we can see that the convergence rate's dependency on $C_1, C_2$ is complicated, and it is hard to find the optimal choice of $C_1$ and $C_2$. However, from our numerical experiments in the appendix (Figure 1.), where we ploted the convergence results with various sets of $C_1, C_2$, it is favorable to choose $C_1 = C_2$ and $C_1, C_2$ both as small as possible.
>
> * Independence of $G, \sigma$: In page 8 after Theorem 3.8, we claim that the result of Theorem 3.8 does not rely on the specific values of $G, \sigma$. Let us clarify such independence. For convenience, we put the choice of $\sigma_1$, the DP noise standard deviation here: $$\sigma_1^2 \geq \frac{32T(C_1^2 + 2\min\{C^2, (\max\{0, G+\sigma -C_1\})^2\})\log(1/\delta)}{N^2\epsilon^2},$$ where we substitute the definition of $\tilde{G}$ and $G'$. Note that the choice of $C_1, C$ are independent of $G, \sigma$. Therefore, in the above bound we have $2\min\{C^2, (\max\{0, G+\sigma-C_1\})^2\} \leq 2C^2$. Therefore, we can choose $$\sigma_1^2 \geq \frac{32T(C_1^2 + 2C^2)\log(1/\delta)}{N^2\epsilon^2},$$ so that the choice of $\sigma_1$ is indeed **independent** of the specific value of $G, \sigma$.
>
> * Effect of the clip norm on the learning rate: Intuitively, using smaller clipping threshold should result in constant slower convergence. However, in our final results, such effect is unclear, as it is merged into the bound of the learning rate $\eta$ ($\mathcal{O}(\frac{\sqrt{C_2}}{\sqrt{T}})$). Smaller $C_2$ results in smaller choice of $\eta$, and smaller $\eta$ results in smaller $\kappa$ and eventually affects the convergence speed of $\mathcal{L}^0$. However, in the **optimized** choice of $\eta = \frac{\log(\mu^2 \mathcal{L}^0T/L(d\sigma_1^2+5\sigma^2/4B+C_1^2/2))}{\mu T}$, the effect of the clipping threshold is minor, since we are choosing learning rate with smaller order (i.e., $O(1/T)$ compared with $O(\frac{\sqrt{C_2}}{\sqrt{T}})$).

---

> > ### Author Response · Authors · 2023-11-23
> > **Response to Reviewer abPz (Part II)**
> >
> > ## Experiments
> >
> > * More experiments: Indeed, we agree with the reviewer that more experiments would strengthen the paper and make it more convincing. Indeed, in this work, we tried our best to strike a balance between demonstrating both theoretical and  practical impact of the proposed algorithm. Unlike some exisitng works such as Varshney et al., 2022, which purely focused on theoretical analysis of a new algorithm, we have conducted extensive experiments on both vision tasks (ViT-small model, on Cifar-10, Cifar-100 datasets) and NLP tasks (GPT model, on E2E NLG Challenge dataset). To our understanding, not many recent works that propose theoretically guaranteed algorithms have verified their algorithms in such a wide choice of datasets/models.
> > Of course, we completely agree with the reviewer that it will be beneficial to test the proposed algorithm on a braoder selection of data sets and models to show its generalizability to different settings. However, due to the space and time limitation of the rebuttal, it is hard for us to conduct more thorough experiments. Nevertheless, we have provided the codes for our algorithm in the supplementary material, and we plan to  opensource the code, together with our benchmarking results so that other researchers can compare the performance of their own algorithms with ours. Meanwhile, on this opensource project, we will continue test our algorithms and providing benchmark results on different datasets/models, as suggested by the reviewer.
> > * How is the clip norm of DPSGD tuned: The clipping thresold of DPSGD are selected following the existing papers (e.g., Alexey Kurakin et al., 2022, Da Yu et al. 2021). By fixing the clipping threshold, we tuned the hyper-parameters of DPSGD ($\eta, B, T$) to achieve the optimal performance.
> >
> > ## Clarity
> > ### Main paper
> > * DPSGD-GC and DPSGD: we thank the reviewer for the advice.
> > * $(\epsilon, \delta)$-DP. Indeed, directly using RDP in the main paper for the results would be straightforward. However, as $(\epsilon, \delta)$-DP would be a more commonly used notation in the previous research. Therefore, we first introduce $(\epsilon, \delta)$-DP in the main results in order to have a direct comparision with DPSGD.
> > *  We agree with the reviewer that $\mathbb{E}_{i}$ should not be used and we will fix that in the revised manuscript. However, $g_i$ is defined in Sec. 2.1 (notations and assumptions) right before eq.(2). $g_i$ is used to avoid the redundant notation of $\nabla f(x;\xi_i)$ and save space only when there is no ambiguity. We will also consider remove the notation of $g_i$ in the revised manuscript.
> >
> >
> > ### Proofs
> > * Definition of $\Delta^t$ and $\alpha^t_e$： we will put a table of symbols used in the paper at the beginning of the appendix in the revised manuscript for clarity.
> > * $\mathcal{A}^t_1, \mathcal{A}^t_2, \mathcal{H}^t$: Let us clarify what these notations refers to: 1) $\mathcal{A}^t_1$ denotes the update and outout of $x^t$ of DiceSGD, i.e, Lines 4 (random sampling),5 (computing $v^t$), and 6 (computing $x^{t+1}$) of Algorithm 2; 2) \mathcal{A}^t_2 denotes the update and output of $e^t$, i.e., Lines 4, 5, and 7 (computing $e^{t+1}$) of Algorithm 2; and $\mathcal{H}^t$ denotes the output sequence of $x^t$, i.e., $\mathcal{H}^t = \\{x^0, x^1, \dots, x^t\\}$. We will add this explanations to the revised manuscript.
> > * Proof of Step II page 18: We cleaned the notations and definitions used in proof step II in our revised manuscript.
> >
> > ## Minors:
> > * We thank the reviewer for pointing out these issues and we have corrected them in the revised manuscript.
> > * We will remove $\kappa$ in the main paper as the reviewer suggested.
> > * For Lemma A.11 we need $\sigma>4C$. In the original paper, the condition is indeed $\sigma>4$. This difference is because that in the original paper,  the default algorithm sensitivity is $1$, i.e., $C=1$. In our Lemma A.11, we explicitly include the dependency on $C$.
> >
> > ## Question
> > * In Step III, $\Delta_g^t$ and $g^t_i$ are defined conditioned on $\mathcal{H}^t = \\{x_1, \dots, x^t\\}$ (i.e., based on the observation of $x^t$'s). Therefore, they are computed based on the same $x^t$ instead of having different $x^t, x'^t$.

---

### Meta-Review · Area_Chair_L5sT · 2023-12-05

**Metareview:**

Summary: The authors provide an error-feedback (EF) DP algorithm to mitigate the bias introduced in norm clipping in DP-SGD, by maintaining the clipped component in a separate accumulator that is also added to the update at each iteration.

Strength: The clipping thresholds can be arbitrarily chosen. In comparison with adaptive clipping, the proposed algorithm assumes a less restrictive problem class. The proposed method outperforms DP-SGD.

Weakness: The analysis is done in convex regimes, while the experiments are in non-convex settings.

**Justification For Why Not Higher Score:**

The scores are not high enough to be spotlight or oral presentations.

**Justification For Why Not Lower Score:**

The paper makes a clear contribution to improving DP-SGD.

---

### Decision · Program_Chairs · 2024-01-16

Accept (poster)